# The final step of 40S ribosomal subunit maturation is controlled by a dual key lock

Laura Plassart[1†], Ramtin Shayan[1†§], Christian Montellese[2#], Dana Rinaldi[1], Natacha Larburu[1¶], Carole Pichereaux[3], Carine Froment[3], Simon Lebaron[1], Marie-Françoise O'Donohue[1], Ulrike Kutay[2], Julien Marcoux[3], Pierre-Emmanuel Gleizes[1‡*], Celia Plisson-Chastang[1‡*]

[1]Molecular, Cellular and Developmental Biology department (MCD), Centre de Biologie Integrative (CBI), University of Toulouse, CNRS, UPS, Toulouse, France; [2]Institut für Biochemie, ETH Zürich, Zurich, Switzerland; [3]Institut de Pharmacologie et Biologie Structurale, Université de Toulouse, CNRS, UPS, Toulouse, France

*For correspondence:
pierre-emmanuel.gleizes@univ-tlse3.fr (P-EG);
celia.plisson-chastang@univ-tlse3.fr (CP-C)

[†]These authors contributed equally to this work
[‡]These authors also contributed equally to this work

Present address: [§]Institute of Structural and Molecular Biology, Birkbeck, University of London, London, United Kingdom; [#]CSL Behring, CSL Biologics Research Center, Bern, Switzerland; [¶]Department of Life Sciences, Imperial College London, London, United Kingdom

Competing interests: The authors declare that no competing interests exist.

**Abstract** Preventing premature interaction of pre-ribosomes with the translation apparatus is essential for translational accuracy. Hence, the final maturation step releasing functional 40S ribosomal subunits, namely processing of the 18S ribosomal RNA 3′ end, is safeguarded by the protein DIM2, which both interacts with the endoribonuclease NOB1 and masks the rRNA cleavage site. To elucidate the control mechanism that unlocks NOB1 activity, we performed cryo-electron microscopy analysis of late human pre-40S particles purified using a catalytically inactive form of the ATPase RIO1. These structures, together with in vivo and in vitro functional analyses, support a model in which ATP-loaded RIO1 cooperates with ribosomal protein RPS26/eS26 to displace DIM2 from the 18S rRNA 3′ end, thereby triggering final cleavage by NOB1; release of ADP then leads to RIO1 dissociation from the 40S subunit. This dual key lock mechanism requiring RIO1 and RPS26 guarantees the precise timing of pre-40S particle conversion into translation-competent ribosomal subunits.

## Introduction

Synthesis of eukaryotic ribosomes relies on a large array of ribosome biogenesis factors (RBFs) that coordinate the multiple steps of pre-ribosomal RNA (pre-rRNA) modification, cleavage, and folding, together with ribosomal protein (RP) assembly (*Bohnsack and Bohnsack, 2019*). Progression through this process, defined by the timely association or dissociation of RBFs and RPs to pre-ribosomal particles and the gradual maturation of pre-rRNAs, is tightly monitored from one stage to the next. These monitoring mechanisms not only ensure quality control along this intricate biosynthetic pathway, but also prevent binding of immature ribosomal subunit precursors (pre-ribosomes) to mRNAs or to components of the translation apparatus. Such interactions would not only interfere with ribosome biogenesis, but also affect translation accuracy. The recent discovery of congenital diseases or cancers linked to ribosome biogenesis defects and translation dysregulation underscores the importance of the control mechanisms that license newly formed ribosomal subunits to enter the translation-competent pool of ribosomes (*Aubert et al., 2018*; *Bohnsack and Bohnsack, 2019*; *Sulima et al., 2017*).

After initial assembly in the nucleolus, pre-40S particles are rapidly transported to the cytoplasm (*Rouquette et al., 2005*), where translation takes place. Cytoplasmic precursors to the 40S ribosomal subunits (pre-40S particles) closely resemble their mature counterparts (*Ameismeier et al., 2018*; *Larburu et al., 2016*), which possess binding sites for numerous components of the

translation machinery including factors of the translation initiation complex, tRNAs and mRNAs. Thus, pre-40S particles would be especially prone to premature interactions with the translation apparatus without the presence of several RBFs (Bystin/ENP1, LTV1, RIO2, TSR1, DIM2/PNO1, NOB1), which occupy the binding sites of translation partners near their 'head' and 'platform' domains (*Larburu et al., 2016*; *Strunk et al., 2012*). These two structural domains undergo several remodeling steps in the cytoplasm leading to the gradual release of these RBFs (*Ameismeier et al., 2018*; *Zemp et al., 2014*; *Zemp et al., 2009*). This maturation process ends with the cleavage of the 18S rRNA 3′ end and the dissociation of the last RBFs, DIM2, and NOB1, which converts pre-40S particles into functional subunits. Interfering with this late stage can result in the incorporation of immature 40S subunits in the translation pool (*Belhabich-Baumas et al., 2017*; *Parker et al., 2019*).

Up to this final stage, the 3′ end of the 18S rRNA is extended with remnants of the internal transcribed spacer 1 (ITS1). In human cells, this last precursor to the 18S rRNA (called 18S-E pre-rRNA) is generated in the nucleolus by endonucleolytic cleavage of earlier pre-rRNAs 78 or 81 nucleotides downstream of the mature 3′ end (*Preti et al., 2013*; *Sloan et al., 2013*; *Tafforeau et al., 2013*). This 3′ tail is then gradually trimmed by exonucleases before and after nuclear export, including PARN in the nucleus (*Montellese et al., 2017*). However, processing of the 18S rRNA 3′ end is finalized by endonuclease NOB1, which cleaves at the so-called site 3. NOB1 is already incorporated into the pre-40S particle in the nucleolus on the platform domain near site 3, but it is restricted from cleaving the rRNA by DIM2, another RBF, which contacts NOB1 in the particle and masks the cleavage site on the RNA (*Ameismeier et al., 2018*; *Larburu et al., 2016*). This conformation also maintains a large gap between the catalytic site of NOB1 and its substrate. Dissociation of NOB1 and DIM2 from the pre-40S particles only occurs at the final processing stage, which produces mature 18S rRNA. In addition, NOB1 occupies the mRNA binding cleft and prevents the association of pre-40S particles with mRNAs until this ultimate maturation step (*Parker et al., 2019*). Thus, NOB1 and DIM2 constitute a critical checkpoint controlling the release of nascent 40S subunits into the translating pool.

The RIO1 ATPase was shown to play a critical function in the last maturation step of the 18S rRNA (*Widmann et al., 2012*), but the molecular mechanism driving the accurate activation of rRNA cleavage by NOB1 remains poorly known. RIO1 associates with pre-40S particles briefly before the final cleavage step. Like RIO2, another ATPase of the same family that intervenes earlier in pre-40S particle maturation, RIO1 adopts different conformations depending on its nucleotide binding state (*Ferreira-Cerca et al., 2014*). The absence of RIO1 or suppression of its catalytic activity impairs both rRNA cleavage and release of NOB1 and DIM2 in yeast and human cells (*Ferreira-Cerca et al., 2014*; *Turowski et al., 2014*; *Widmann et al., 2012*), suggesting that RIO1 association to pre-40S particles as well as its ATPase activity are both required to yield functional small ribosomal subunits.

Ribosomal protein RPS26/eS26, which is in close contact with the 18S rRNA 3′ end on the mature small ribosomal subunit, was also shown to be necessary for efficient cleavage by NOB1 in yeast and in human cells (*O'Donohue et al., 2010*; *Peña et al., 2016*; *Schütz et al., 2018*), but its interplay with RIO1, DIM2, and NOB1 remains unclear. Indeed, the precise timing of association of this ribosomal protein to pre-ribosomal particles is a matter of debate. Studies performed in yeast cells suggested that Rps26 is imported into the nucleus and is dissociated from its importin by the RBF Tsr2 in a non-canonical way (*Schütz et al., 2014*). Tsr2 was proposed to then chaperone the incorporation of Rps26 into nucleolar pre-ribosomal particles (*Peña et al., 2016*; *Schütz et al., 2018*). However, RPS26 has not been found so far in cryo-electron microscopy (cryo-EM) structures of nucle(ol)ar small ribosomal subunit precursors in yeast or human, and was only detected in late cytoplasmic ones (*Ameismeier et al., 2020*; *Ameismeier et al., 2018*). Consistently, western blot and proteomics analysis failed to clearly identify RPS26 in nuclear or in early cytoplasmic human pre-40S particles (*Larburu et al., 2016*; *Wyler et al., 2011*). One cannot exclude that association of RPS26 to nuclear precursors might be highly labile and becomes more stable only during later maturation steps, but evidence is still lacking to date for the presence of human RPS26 in early precursors.

Here, we have used cryo-EM to solve the structures of human pre-40S particles trapped with a catalytically inactive form of RIO1 in this final maturation stage. Image analysis revealed two distinct structural states, prior and after 18S rRNA cleavage. In the pre-cleavage state, DIM2 and NOB1 are still in place, while the postion of RIO1 is not clearly defined and RPS26 is absent. In contrast, the post-cleavage state displays RIO1 and RPS26 stably associated with 40S particles, which contain mature 18S rRNA. In vivo assays confirmed the central role of RPS26 for triggering both processing

of the 18S-E pre-rRNA and the release of DIM2, NOB1, and RIO1. In vitro, cleavage of the 18S-E pre-rRNA was partially stimulated by ATP binding to RIO1 in RPS26-depleted pre-40S particles and was enhanced by adding back purified RPS26. These data suggest a model in which ATP-bound RIO1 and RPS26 cooperatively displace DIM2 to activate the final cleavage of the 18S rRNA 3′ end by NOB1.

## Results

### Pre-40S particles purified with a catalytically deficient form of RIO1 display pre- and post-18S rRNA processing structural states

In order to characterize the final cytoplasmic maturation steps undergone by human pre-40S particles, we purified pre-40S particles from a human cell line overexpressing a tagged version of a catalytically inactive form of RIO1 mutated on aspartic acid 324 (D324A) (*Widmann et al., 2012*). Autophosphorylation of this mutant in the presence of ATP was previously shown to be strongly impaired (*Widmann et al., 2012*), and free Pi release activity reduced by at least 50% in its *Chaetomium thermophilum* D281A ortholog (*Ferreira-Cerca et al., 2014*). Protein and RNA composition of the affinity-purified complexes, hereafter called RIO1(kd)-StHA pre-40S particles, were analyzed using SDS–PAGE, northern blot, and bottom-up proteomics (*Figure 1*). SDS–PAGE revealed a complex protein pattern with a typical trail of low-molecular-weight bands corresponding to RPs. Northern blot showed that the 18S-E pre-rRNA was the only 18S rRNA precursor precipitated by this bait, while no rRNA of the large ribosomal subunit could be detected (*Figure 1—figure supplement 1*). Bottom-up proteomics confirmed that the catalytically dead version of RIO1 associates with the methylosome, a complex composed of proteins PRMT5 and MEP50 that methylates proteins involved in gene expression regulation (*Guderian et al., 2011*), in addition to late pre-40S particles (*Montellese et al., 2017*; *Widmann et al., 2012*). Of note, these purified pre-40S particles contain all RPs of the small subunit (RPSs), including the late binding RPS10 and RPS26, but only a handful of ribosome biogenesis factors, namely RIO1 (the bait), DIM2, NOB1, and TSR1 (*Figure 1*, *Figure 1— figure supplement 2*). Other proteins also co-purified with RIO1(kd)-StHA: while some can unambiguously be identified as components of the translation apparatus (eIF4A, RPLs, etc.), it is not known whether others belong to the methylosome or to (pre-)ribosomal particles.

We then performed cryo-EM and single-particle analysis on the particles purified using RIO1(kd)-StHA as bait. 2D classification assays performed with RELION (*Scheres, 2012*) yielded class sums corresponding to pre-40S particles as well as various views of the methylosome (*Timm et al., 2018*; *Figure 2*, *Figure 2—figure supplement 1*). Pre-40S views were selected for further processing; an extensive 3D classification scheme resulted in two distinct 3D structures, hereafter called state A and state B, that likely reflect two successive pre-40S maturation steps (*Figure 2*, *Figure 2—figure supplement 1*). Both structures, which represented ~21% (state A) and ~57% (state B) of the pre-40S particles comprised within the analysis, were refined to overall resolutions of 3.2 and 3.0 Å, respectively, according to RELION's gold-standard FSC (*Figure 2—figure supplements 1* and *2*).

State A harbored an rRNA scaffold closely resembling that of the mature 18S rRNA. On the head region, the three-way junction formed by rRNA helices h34, h35, and h38 is fully formed (*Mohan et al., 2014*), while RPS3, RPS10, and RPS12 occupy their final position. Like in many other pre-40S structures (*Ameismeier et al., 2018*; *Heuer et al., 2017*; *Mitterer et al., 2019*; *Scaiola et al., 2018*), the upper part of rRNA h44, located on the intersubunit side, appears detached from the body (*Figure 3a*). Low-pass filtering of the cryo-EM map of state A also revealed the presence of TSR1 in this region (*Figure 3—figure supplement 1*). Combined with the bottom-up proteomic analysis that revealed the presence of this RBF among the co-purified proteins, this suggests that TSR1 might still be loosely and/or flexibly bound to pre-40S particles at this maturation state. Of note, the presence of TSR1 in late pre-40S particles and mature 40S subunits was recently observed by others (*Thoms et al., 2020*), reinforcing the idea that TSR1 could stay associated with small ribosomal subunits until their entry into translation initiation. On the platform region, the cryo-EM density map allowed to unambiguously position nucleotide A1870, belonging to the ITS1 after the 3′ end of the mature 18S rRNA. In this state, DIM2 protects this region of the 18S-E pre-rRNA and impedes endonucleolytic cleavage by NOB1, located right next to DIM2 (*Figures 2* and *3b*). These results suggest that RIO1(kd)-StHA pre-40S particles in structural state A are in an

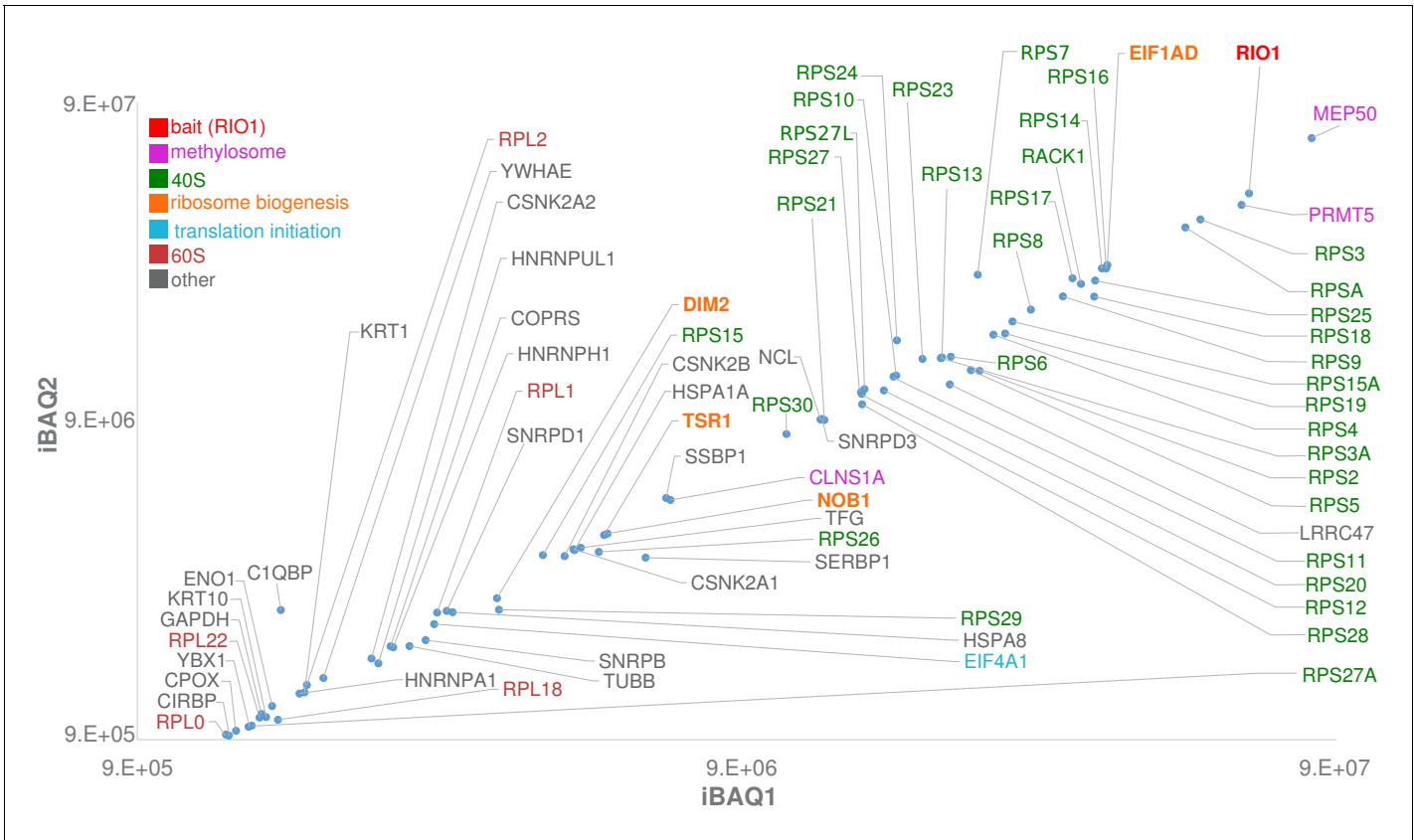

**Figure 1.** Label-free bottom-up proteomic analysis of RIO1(kd)-StHA co-purified proteins. The most intense proteins (first two logs) with an observed/observable peptide ratio > 30% are displayed and color coded as indicated on the graph. Three independent experimental replicates were performed. This plot represents iBAQs (intensity-based absolute quantification) of experimental replicate 1 (iBAQ1) against experimental replicate 2 (iBAQ2). The iBAQ value is obtained by dividing protein intensities by the number of theoretically observable tryptic peptides (*Schwanhäusser et al., 2011*). Other plots are displayed in *Figure 1—figure supplement 2*. Of note, USP16, which was recently identified as a key player in both ribosome biogenesis and translation (*Montellese et al., 2020*), was also identified among RIO1(kd)-StHA proteic partners. It ranked at the 62th position when quantifying proteins based on their normalized abundances and at the 98th position when quantifying proteins based on their iBAQs (see the PRIDE repository [*Perez-Riverol et al., 2019*]; dataset identifier: PXD019270).

The online version of this article includes the following figure supplement(s) for figure 1:

**Figure supplement 1.** Purification of RIO1(kd)-StHA-containing particles.

**Figure supplement 2.** Label-free bottom-up proteomic analyses of RIO1(kd)-StHA co-purified proteins.

immature state in which the rRNA scaffold harbors a quasi-mature conformation, but the remaining nucleotides of the ITS1 have not yet been cleaved off by NOB1. Furthermore, NOB1 and DIM2 are the only two RBFs that can clearly be distinguished, while RIO1(kd)-StHA, the protein used as purification bait, cannot be clearly positioned on this cryo-EM map. This suggests that RIO1(kd)-StHA is not structurally stabilized onto state A pre-40S particles. Indeed, while this manuscript was under revision, a cryo-EM study focusing on similarly late human pre-40S particles revealed that RIO1 can adopt several positions in maturing pre-40S particles (*Ameismeier et al., 2020*).

On the contrary, the cryo-EM map corresponding to state B harbored a density located on the intersubunit side, at the back of the head region, into which the X-ray structure of RIO1 could unambiguously be fitted (*Ferreira-Cerca et al., 2014*). Of note, RIO1 occupies a position strongly overlapping with that of RIO2 in earlier pre-40S maturation states (*Ameismeier et al., 2018*; *Figure 3—figure supplement 1*). This observation explains why both ATPases are not found together in pre-40S particles and confirms that RIO1 replaces RIO2 at the back of the head (*Ameismeier et al., 2018*; *Knüppel et al., 2018*; *Widmann et al., 2012*). The C-terminal domain of RIO2 was shown to be deeply inserted within the body of human pre-40S particles (*Ameismeier et al., 2018*). In contrast, the C-terminal domain of RIO1 is not resolved here; furthermore, the upper part of rRNA helix

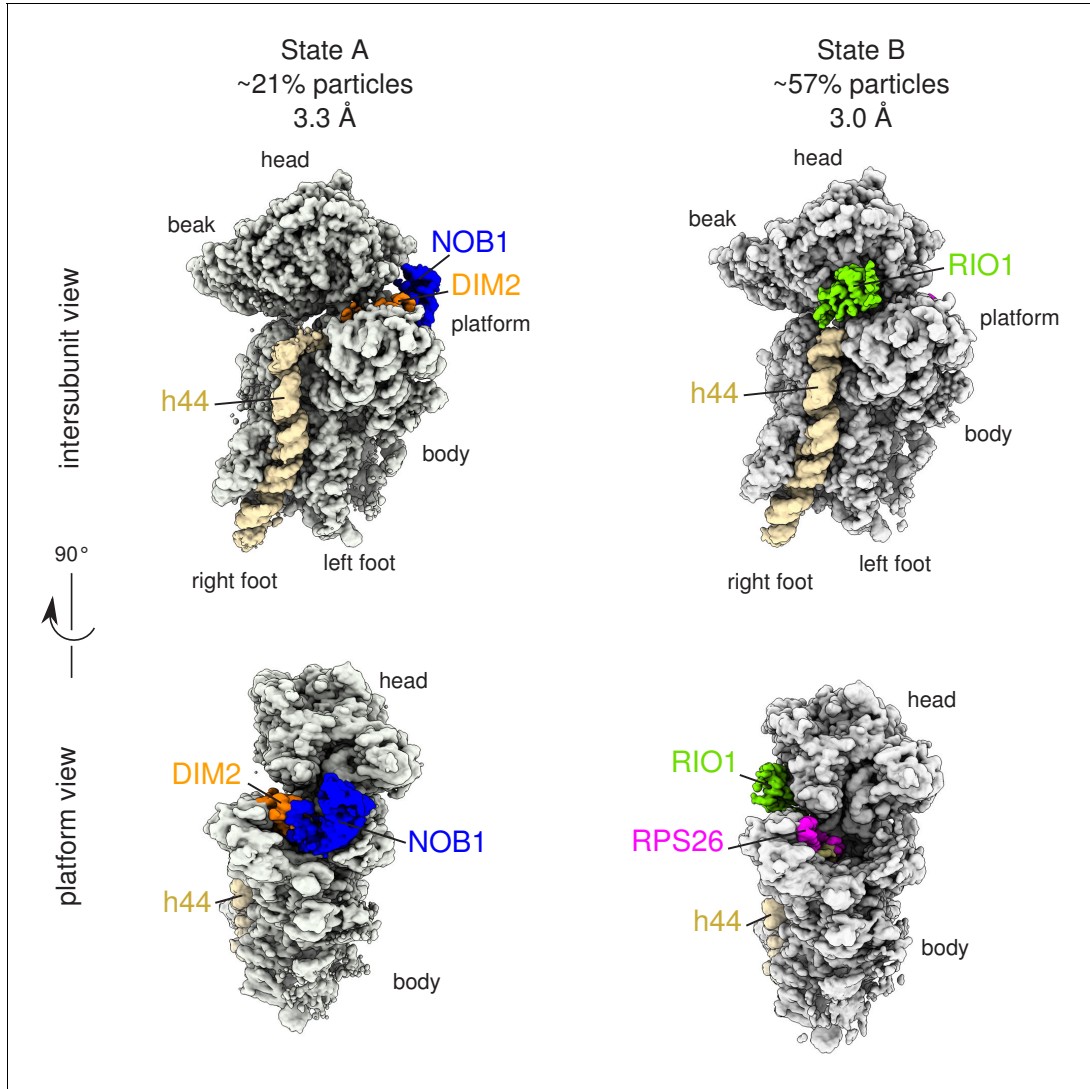

**Figure 2.** Cryo-EM and single-particle analysis reveal two distinct structural states, pre- (state A) and post- (state B) 18S-E rRNA cleavage. Surface views of cryo-EM maps of RIO1(kd)-StHA pre-40S particles in structural state A (left) and state B (right). Ribosomal proteins, rRNA segments, and RBFs of interest have been segmented and colored as indicated on the figure. Image processing details are shown in *Figure 2—figure supplements 1* and *2*. The online version of this article includes the following figure supplement(s) for figure 2:

**Figure supplement 1.** Cryo-EM image processing scheme.

**Figure supplement 2.** Details of the cryo-EM structures/model validation.

h44 occupies its mature position, which would preclude insertion of the C-terminal domain of RIO1 at the same position as the C-terminal domain of RIO2. Indeed, the strong sequence divergence of the C-termini of RIO proteins suggests that these domains play a major role in the functional specificity of these proteins, which is supported by the structural difference observed here.

As previously observed in the structure of wild-type RIO1 solved by X-ray crystallography (*Ferreira-Cerca et al., 2014*), our cryo-EM map revealed that the catalytic pocket of mutant RIO1 encloses an ADP together with phospho-aspartate pD341 (*Figure 3c*, *Figure 3—figure supplement 2*). Thus, the D324A RIO1 mutation does not fully prevent ATP hydrolysis, but rather blocks the release of its reaction products. ATP hydrolysis is thought to be accompanied by a significant conformational change of RIO1 (*Knüppel et al., 2018*; *Kühlbrandt, 2004*), which might be essential for stable association of RIO1 to pre-40S particles. Blocking of RIO1 in this conformation is likely to explain why this point mutation traps the protein on the particles. Furthermore, the RIO1(kd) catalytic pocket appears to be closed through a pi-stacking interaction between phenylalanine F328 and 18S rRNA

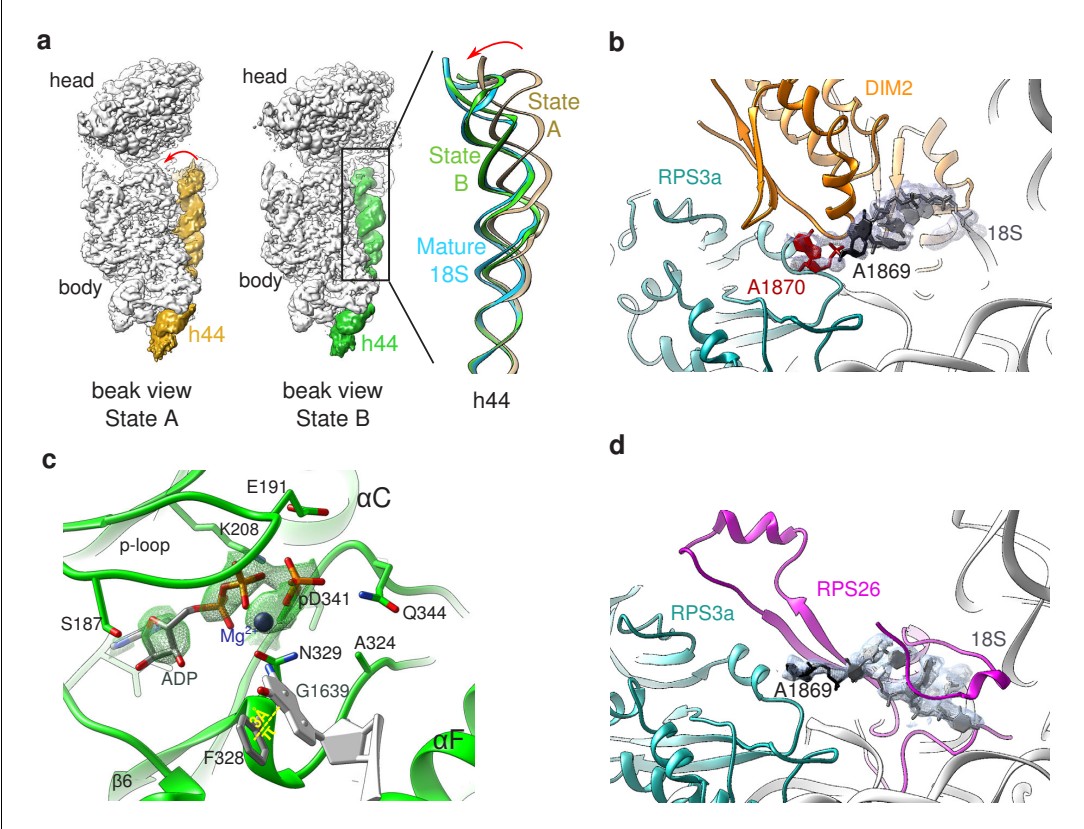

**Figure 3.** Structural details of RIO(kd)-StHA pre-40S particles. (**a**) The upper part of 18S rRNA helix h44 is in an immature position in structural state A. The cryo-EM density map corresponding to this helix has been segmented (beige density); the atomic model of rRNA h44 in structural state A is represented in golden; superimposed 18S rRNA h44 as found in structural state B and in the mature 40S subunit (PDB 6EK0 *Natchiar et al., 2018*) are in green and blue, respectively. (**b**) Close-up on the platform domain of structural state A. Segmented cryo-EM density corresponding to 18S-E pre-rRNA is shown as a grey mesh. The 3'-end of the mature 18S rRNA (nucleotides 1865–1869) is shown in black, while A1870 in the ITS1 is in red. The 18S rRNA is otherwise shown as a gray ribbon. DIM2 is in orange; RPS3a is in turquoise. NOB1 was removed from this representation for the sake of clarity. (**c**) The catalytic pocket of RIO1 is in an 'active' state within structural state B pre-40S particles, and carries an ADP and a phospho-aspartate (pD341). The cryo-EM density corresponding to ADP, p-Asp, and magnesium is shown as a green mesh. RIO1 is shown in green; ADP in dark gray, $Mg^{2+}$ in dark violet, and 18S rRNA G1639 closing RIO1 catalytic domain by a pi-stacking interaction (yellow dashed line) with RIO1 Phe 328 in light gray. (**d**) Close-up on the platform domain of structural state B. Segmented cryo-EM density corresponding to 18S rRNA 3'-end is shown as a light gray mesh. 18S rRNA 3'-end (nucleotides 1865–1869) is shown in black, while otherwise as gray ribbon. RPS26 is in magenta and RPS3a in turquoise.

The online version of this article includes the following figure supplement(s) for figure 3:

**Figure supplement 1.** Details of the structural analysis of hRIO1(kd)-StHA pre-40S particles.

**Figure supplement 2.** RIO1(kd) carries an hydrolyzed ATP in its catalytic site.

G1639 (*Figure 3c*). This highly conserved nucleotide plays a crucial role in tRNA translocation, which puts RIO1 in a good position to probe this mechanism (see Discussion).

In state B, the rRNA scaffold harbors a fully mature conformation. Contrary to what was observed on the platform in state A, no density corresponding to nucleotides belonging to the ITS1 could be detected, suggesting that the 18S rRNA 3' end is mature. Furthermore, neither NOB1 nor DIM2 could be found in this area; instead, a cryo-EM density sheathing the 18S rRNA 3' end revealed the presence of RPS26 (*Figure 3d*). These observations indicate that state B corresponds to particles after rRNA cleavage by NOB1. This conclusion was further supported by the analysis of the 3'-end of the 18S rRNA by RNase H digestion assays (*Figure 4a*). As expected, early and intermediate cytoplasmic pre-40S particles purified using tagged forms of LTV1 or DIM2 as baits (*Ameismeier et al., 2018*; *Wyler et al., 2011*) contained a large majority of 18S-E pre-rRNAs. Most of the 18S-E precursors in these particles included 4 to 9 nucleotides of the ITS1 (*Figure 4*, *Figure 4—figure*

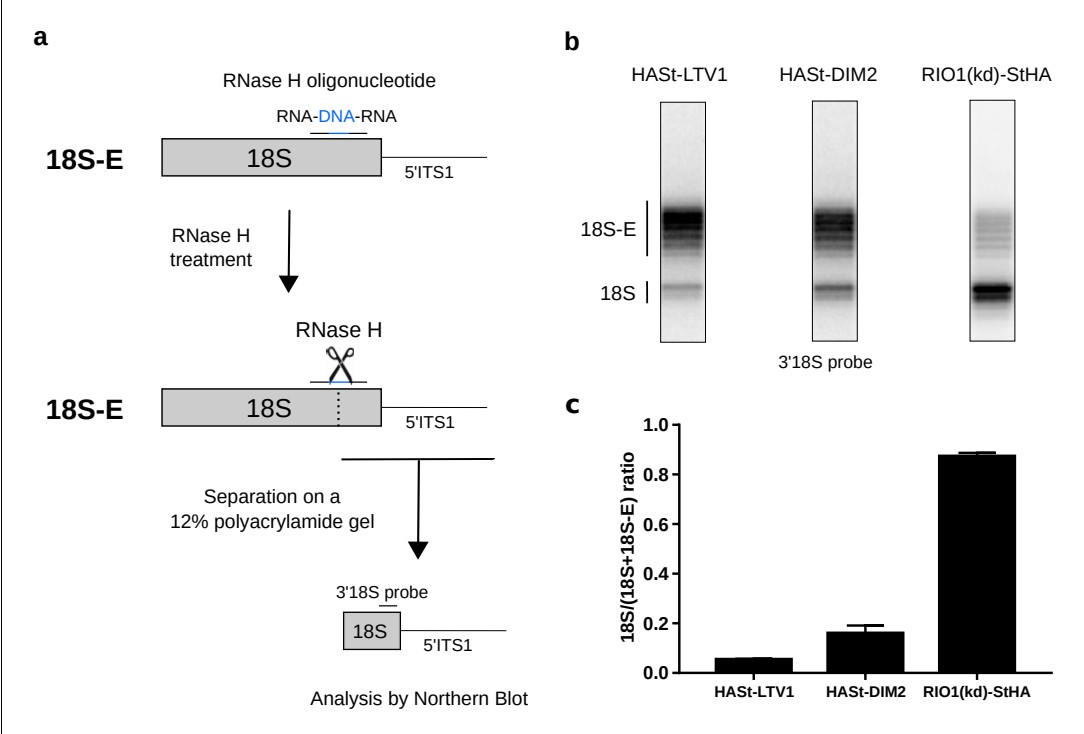

**Figure 4.** Late RIO1(kd)StHA pre-40S particles contain a high proportion of mature 18S rRNA. (a) Diagram representing steps of the pre-40S rRNA digestion by RNase H. (b) RNase H assays were performed on RNAs extracted from pre-40S particles purified with the mentioned StHA-tagged bait and separated on a 12% polyacrylamide gel. The 18S rRNA and its precursors were revealed by the 3'18S radiolabeled probe. Bands are separated with single nucleotide resolution, as shown in *Figure 4—figure supplement 1*. (c) Signals corresponding to the 18S-E and 18S rRNAs were quantified by phosphorimaging and represented by the 18S/(18S + 18S-E) ratio for the different purified pre-40S particles. The average of three independent experiments is shown, with the standard deviation indicated on top of the histogram.

The online version of this article includes the following figure supplement(s) for figure 4:

**Figure supplement 1.** Assessment of the size of the ITS1 in purified pre-40S particles.

supplement 1). In stark contrast, this experiment showed that RIO1(kd)-StHA pre-40S particles mostly contained mature 18S rRNA (*Figure 4b,c*).

We conclude that states A and B of the pre-40S particles here isolated with RIO1(kd)-StHA correspond to two very late maturation stages, just before and right after cleavage of the 18S rRNA 3′ end. These data indicate that rRNA cleavage and release of NOB1 and DIM2 coincide with the association of RIO1 in a stable conformation within the pre-40S particle and the incorporation of RPS26 in its final location. RIO1 enzymatic activity, and more specifically the release of the ATP hydrolysis products, seems required for its own dissociation from the particle rather than for the 18S-E pre-rRNA processing by NOB1.

## RPS26 is required for efficient rRNA cleavage and release of NOB1 and DIM2

States A and B of the RIO1(kd)-StHA pre-40S particles harbor two distinct conformations of the platform domain: in state A, the 18S-E pre-rRNA 3'-end is protected by DIM2, which prevents endonucleolytic cleavage by NOB1; in state B, the platform displays a mature conformation, with RPS26 replacing DIM2. Both RPS26 and RIO1 were shown to be necessary for processing of the 18S-E pre-rRNA to occur in human cells (*O'Donohue et al., 2010*; *Widmann et al., 2012*), which argues for the two proteins being involved in transition from state A to state B. Accommodation of RPS26 in its mature position requires prior release of DIM2, which occupies the binding site of RPS26.

In order to assess the role of RPS26 binding to pre-40S particles in 18S rRNA maturation as well as DIM2 and NOB1 release, we analyzed the RNA and protein composition of intermediate and late

pre-40S particles purified from cells depleted of RPS26 using a specific siRNA (siRPS26). The level of 18S-E pre-rRNA cleavage in pre-40S particles was measured by RNase H analysis in pre-40S particles isolated using different baits (RIO1(kd)-StHA or RIO1(wt)-StHA, HASt-LTV1) (*Figure 5a,b*). The amount of 18S-E pre-rRNA within early cytoplasmic pre-40S particles purified with HASt-LTV1 remained unchanged upon RPS26 depletion (*Figure 5a*), consistent with the absence of RPS26 in these particles (*Ameismeier et al., 2018*; *Larburu et al., 2016*; *Wyler et al., 2011*). In contrast, we observed a strong defect of 18S-E pre-rRNA processing in RIO1(kd)-StHA particles upon knockdown of RPS26, as evidenced by a ~60% decrease of the 18S/(18S + 18S-E) ratio when compared to the same particles isolated from cells treated with scramble siRNA. Similar observations were obtained when purifying pre-40S particles with a wild-type version of RIO1 as bait (RIO1(wt)-StHA pre-40S particles). These experiments indicate that the absence of RPS26 impedes 18S-E pre-rRNA maturation and that the action of RPS26 in 18S-E pre-rRNA cleavage takes place within RIO1-containing late cytoplasmic pre-40S particles.

We then used western blot analyses to monitor how RPS26 depletion influences the release of NOB1 and DIM2 (*Figure 5c*). As expected from previous studies (*Larburu et al., 2016*; *Wyler et al., 2011*), we did not detect RPS26 in early cytoplasmic HASt-LTV1 pre-40S particles. Consistently, knockdown of RPS26 did not influence the presence of NOB1 and DIM2 in these early cytoplasmic pre-40S particles, as attested by measuring the NOB1/RPS19 or DIM2/RPS19 ratios (*Figure 5d*), nor did it affect the dissociation of ENP1 and RIO2. In contrast, we observed a strong increase of NOB1 and DIM2 levels relative to RPS19 in RIO1(kd)-StHA pre-40S particles purified from RPS26-depleted cells when compared to control cells (*Figure 5c,d*), which suggests that the absence of RPS26 prevents the release of NOB1 and DIM2 and traps the particles in state A. This hypothesis was further supported when we analyzed pre-40S particles purified using wild-type RIO1 as bait (RIO1(wt)-StHA). We have previously shown that RIO1(wt)-StHA does not co-purify efficiently with pre-40S particles when compared to the catalytically inactive version of RIO1 (*Montellese et al., 2020*; *Widmann et al., 2012*), which was confirmed here by the very low amount of both pre-40S RBFs and RPS19 detected by western blot (*Figure 5c*). However, upon RPS26 depletion, RIO1(wt)-StHA co-purified with late cytoplasmic pre-40S particles, as attested by the presence of NOB1 and DIM2 (*Figure 5c*), as well as that of 18S-E pre-rRNA (*Figure 5a*). These data indicate that RIO1 association with late cytoplasmic pre-40S particles is stabilized in the absence of RPS26, while the release of NOB1 and DIM2 is inhibited.

Comparative proteomics analyses of RIO1(kd)-StHA-associated pre-40S particles in the absence or presence of RPS26 confirmed that no other RPS was lost upon RPS26 depletion (*Figure 5—figure supplement 1*). This experiment also revealed a significant decrease of EIF1AD co-purification with RIO1(kd)-StHA upon RPS26 depletion (log 2 (fold change) = −2.1, corresponding to a 4.2-fold decrease; p-value=8.9E-3). EIF1AD was recently shown to be a new RBF present in late pre-40S particles in direct contact with RIO1 (*Ameismeier et al., 2020*). But whether RPS26 depletion affects EIF1AD association to RIO1 within or out of pre-40S particles cannot be distinguished by this experiment. To assess incorporation of EIF1AD in RIO1(kd)-StHA pre-40S particles, we evaluated the ratio of EIF1AD relative to RPS19 within pre-40S particles by western blot and observed a mild 1.3-fold reduction upon RPS26 depletion (*Figure 5—figure supplement 2*). As a comparison, the levels of NOB1 or DIM2 relative to RPS19 increased by a factor of 1.6 and 2, respectively (*Figure 5c,d*, *Figure 5—figure supplement 2*). These results indicate that loss of RPS26 does not strongly impact the presence of EIF1AD onto RIO1(kd)-StHA pre-40S particles. The stronger decrease observed by mass spectrometry may be also related to EIF1AD association with RIO1(kd)-StHA out of the pre-40S particles and be an indirect effect of RPS26 depletion. Altogether, these data lead us to conclude that RPS26 intervenes directly in the mechanism triggering rRNA cleavage by NOB1 at site 3 and dissociation of NOB1, DIM2, and RIO1 from pre-40S particles.

## ATP binding by RIO1 and addition of RPS26 stimulate in vitro rRNA cleavage by NOB1 in RPS26-depleted pre-40S particles

RIO protein kinases have been proposed to act as conformation-sensing ATPases rather than kinases (*Ferreira-Cerca et al., 2014*). Previous studies in the yeast *Saccharomyces cerevisiae* have shown that rRNA cleavage by Nob1 in cytoplasmic pre-40S particles purified with tagged Rio1 was stimulated in vitro by ATP binding to Rio1 (*Turowski et al., 2014*). The D324A point mutation introduced in RIO1(kd)-StHA does not hamper fixation of ATP (*Widmann et al., 2012*), but the structure of

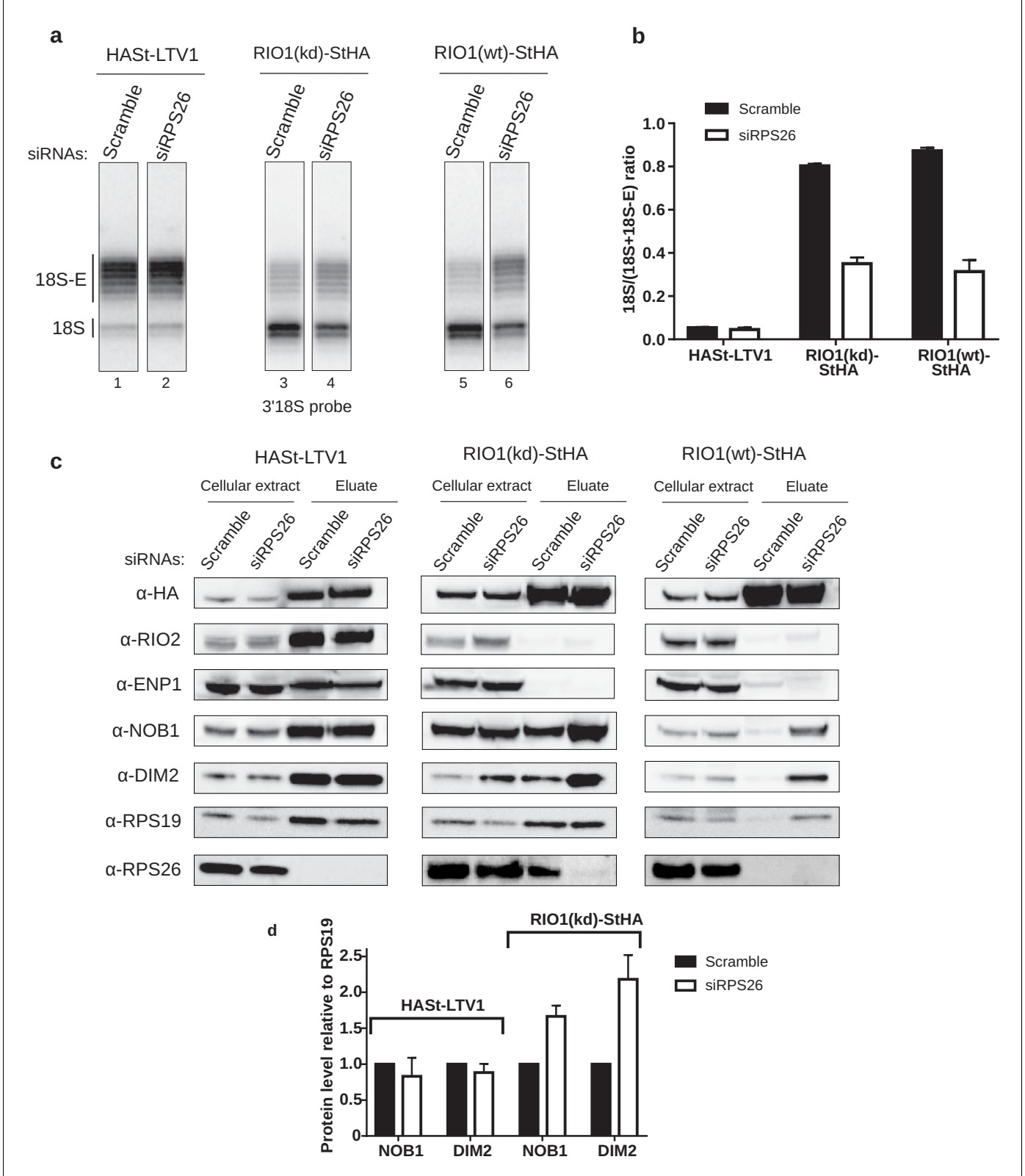

**Figure 5.** RPS26 is required for rRNA cleavage at site 3 as well as NOB1 and DIM2 release. HEK cell lines expressing tagged version of LTV1, the catalytically inactive RIO1-D324A (RIO1 (kd)) or wild-type RIO1 (RIO1(wt)) were treated with scramble or RPS26 siRNAs for 48 hr. (**a**) RNase H assays were conducted as in *Figure 4* on rRNAs of pre-40S particles purified with the mentioned StHA-tagged bait, either from RPS26-depleted or from control cells (scramble siRNA). (**b**) Signals corresponding to the 18S-E and 18S rRNA detected in (**a**) were quantified and represented as the 18S/(18S + 18S-E)

*Figure 5 continued on next page*

*Figure 5 continued*

ratio for the different pre-40S particles. Error bars, s.d. (n = 3) (c) Cell extracts and purified particles were analyzed by western blot using the indicated antibodies. (d) Bands corresponding to DIM2 and NOB1 (in the eluates) were quantified, corrected for pre-40S particle loading (using RPS19) and normalized to the control condition (set to 1). Error bars, s.d. (n = 3).

The online version of this article includes the following figure supplement(s) for figure 5:

**Figure supplement 1.** Comparative proteomics of RIO1(kd)-StHA particles composition upon RPS26 depletion.
**Figure supplement 2.** Western blot analysis of EIF1AD association to pre-40S particles upon depletion of RPS26.

RIO1(kd) in state B shows that the hydrolysis products are trapped in the catalytic site (*Figure 3c*). However, this defective catalytic activity does not block maturation of the 3'-end of 18S rRNA within RIO1(kd)-StHA pre-40S particles. This suggests that ATP binding to the human RIO1 catalytic site is sufficient for 18S-E pre-rRNA cleavage, similar to what was shown in yeast. In order to check whether ATP binding to RIO1 favors rRNA cleavage by NOB1, we purified RIO1(wt)-StHA and RIO1(kd)-StHA human pre-40S particles from RPS26 depleted cells to enrich pre-cleavage state A and performed in vitro cleavage assays.

Based on the conditions established in yeast (*Lebaron et al., 2012*), we added either ATP or AMP-PNP (a non-hydrolysable analog of ATP) to purified pre-40S particles and monitored 18S rRNA 3'-end maturation using RNase H digestion. As shown in *Figure 6* , 18S–E pre-rRNA cleavage was stimulated both by ATP and AMP-PNP in particles purified with RIO1(wt)-StHA or with RIO1(kd)-StHA, albeit a bit more efficiently by ATP as previously observed in yeast (*Figure 6a,b*). This result reinforces the idea that, like in yeast, ATP binding to RIO1 stimulates 18S-E cleavage at site 3 (*Ferreira-Cerca et al., 2014*; *Turowski et al., 2014*). Given the position of RIO1 in the vicinity of DIM2 and NOB1 revealed by the cryo-EM analysis, RIO1 might favor DIM2 displacement and subsequent activity of NOB1 through an ATP-driven local conformational change. Indeed, superimposition of state A and state B atomic models showed steric hindrance between the C-terminal regions of DIM2 and RIO1 (*Figure 6—figure supplement 1*); similarly, putative clashes between RIO1 and DIM2 were also seen by others (*Ameismeier et al., 2020*). These structural observations support a mechanism of competition between RIO1 and DIM2.

We next sought to assess the role of RPS26 and produced recombinant human RPS26 (*Figure 6—figure supplement 2*). Addition of RPS26 to the pre-40S particles depleted of RPS26 stimulated cleavage by NOB1 to the same extent as addition of ATP (*Figure 6a,c*). Importantly, the addition of RPS26 together with ATP significantly increased the fraction of cleaved 18S rRNA when compared to single addition of one or the other (*Figure 6a,c*). These data show that in vitro cleavage by NOB1 in pre-40S particles is stimulated by both RIO1 and RPS26 and support the hypothesis of a cooperative action of RPS26 and RIO1 in triggering the final step of 18S rRNA maturation.

## Discussion

A previous cryo-EM analysis of human pre-40S particles purified using DIM2 as bait revealed several successive structural states, in which the latest one carried NOB1, DIM2, TSR1, as well as a long alpha-helix attributed to LTV1 (*Ameismeier et al., 2018*). Here, by purification of a pre-40S particle via a catalytically inactive form of RIO1, we have uncovered two later structural states in which only the last RBFs found in pre-40S particles, namely NOB1, DIM2, and RIO1, were clearly detected. In addition, one of these states is posterior to 18S-E pre-rRNA cleavage at site 3, as indicated by detection of the mature 3' end of the 18S rRNA, presence of RPS26, and absence of NOB1 and DIM2. These structures arguably correspond to the ultimate maturation stages of the pre-40S particles. Our results point towards a coordinated action of RIO1 and RPS26 in triggering the last 18S rRNA processing step by NOB1, following a sequence represented in *Figure 7* and discussed below.

### The cooperative action of RIO1 and RPS26 unlocks 18S-E rRNA cleavage

Our results show that RIO1 binds to pre-40S particles in the same region as RIO2, which suggests that recruitment of RIO1 may simply follow the release of RIO2. The absence of a clear density for

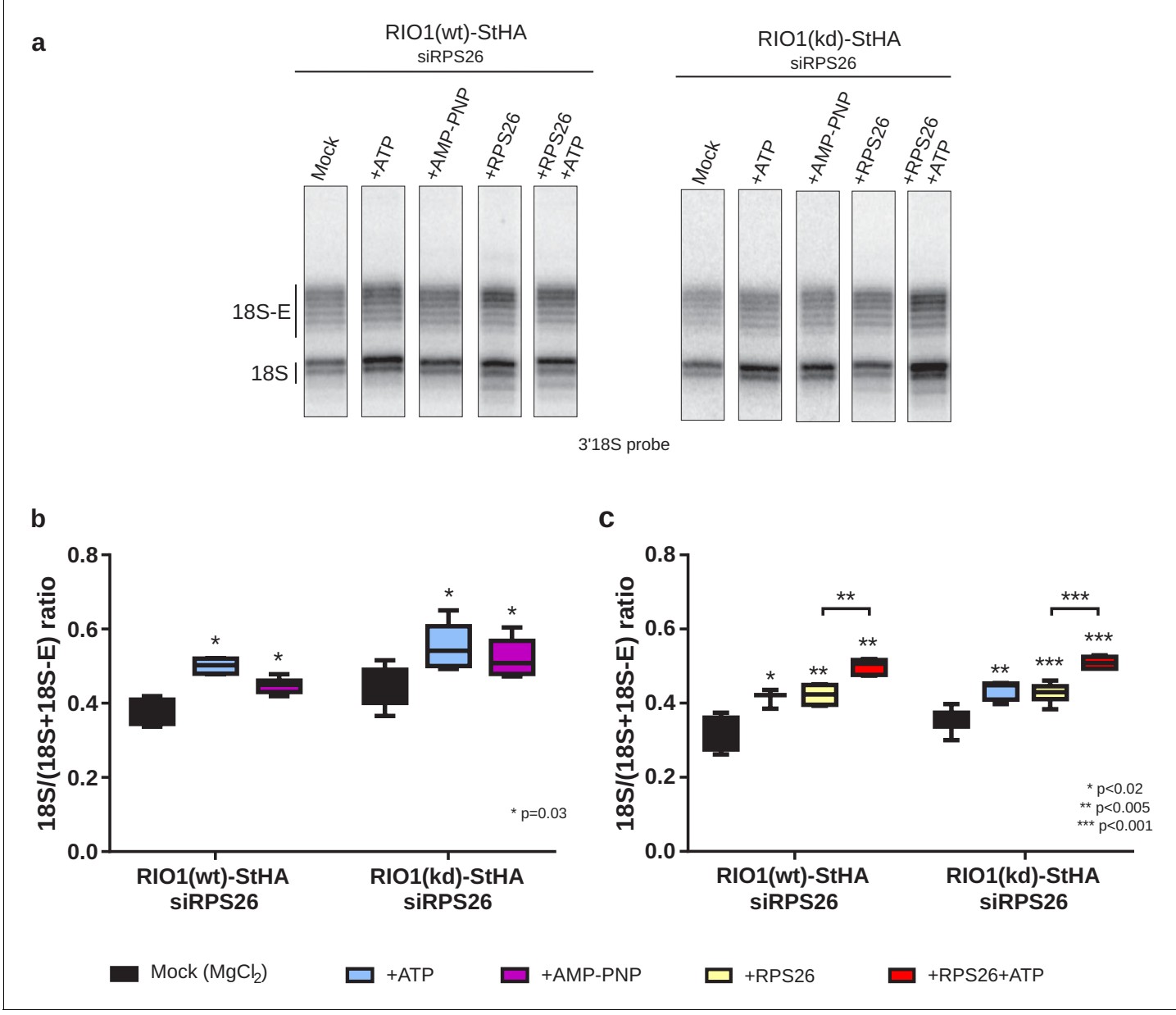

**Figure 6.** In vitro cleavage of the 18S-E pre-rRNA within pre-40S particles is stimulated by ATP addition. HEK cell lines expressing tagged versions of wild-type RIO1 'RIO1(wt)' or of the catalytically inactive 'RIO1(kd)' were treated with RPS26 siRNAs for 48 hr to enrich particles in state A. Pre-40S particles were purified and incubated for 1 hr in buffer alone (mock condition) or in the presence of either 1 mM ATP, 1 mM AMP-PNP, 2 µg of RPS26, or 2 µg of RPS26 plus 1 mM ATP. (**a**) RNAse H assays were performed on the RNAs extracted from the particles. (**b**) The variation of cleavage efficiency upon addition of ATP or AMP-PNP is indicated by the 18S/(18S + 18S-E) ratio. The data correspond to five independent experiments. Statistical analysis was performed with a unilateral paired Wilcoxon test ('sample greater than mock') indicating p-values of 0.031 for all samples. (**c**) The variation of cleavage efficiency upon addition of ATP and/or RPS26 is indicated by the 18S/(18S + 18S-E) ratio. The data correspond to three to six independent measurements for each point. Statistical analysis was performed with a unilateral unpaired Wilcoxon test ('sample greater than mock' or '+RPS26+ATP greater than +RPS26').

The online version of this article includes the following figure supplement(s) for figure 6:

**Figure supplement 1.** Superimposition of atomic models of states A and B reveals overlapping distances (gray lines) between atoms of Proline 351 from RIO1 (green) and of Arginine 247 from DIM2 (orange).

**Figure supplement 2.** Purification of recombinant hRPS26-His.

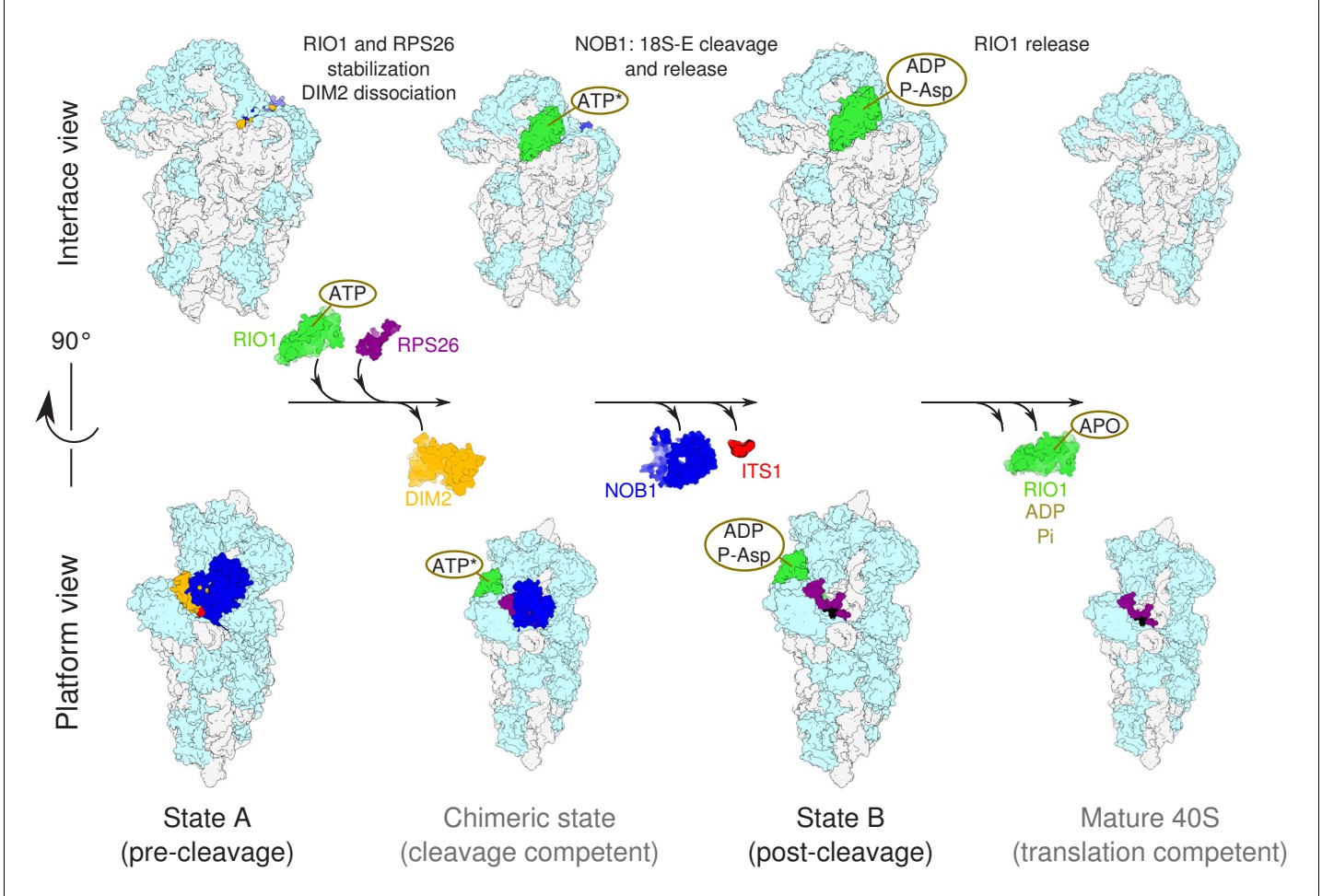

**Figure 7.** Model of the pre-40S last maturation steps triggered by RIO1 and RPS26. Upper and lower panels represent interface and platform views of the small ribosomal subunit, respectively. A putative ITS1 cleavage-competent state is shown to illustrate the transition between the pre-cleavage (state A) and post-cleavage (state B) structures that were resolved in this study. Status of ATP hydrolysis within RIO1 for this maturation state is not known, and thus marked as 'ATP*'.

RIO1 in pre-cleavage state A suggests that RIO1 is initially flexibly associated, while DIM2 and NOB1 are stably bound. A recent cryo-EM study performed on human pre-40S particles revealed significant changes in RIO1 position before 18S-E pre-rRNA cleavage (*Ameismeier et al., 2020*). We hypothesize that we could not precisely position RIO1 in the pre-cleavage state A herein described because our cryo-EM analysis was performed on a significantly smaller scale.

In contrast, state B shows RIO1 in a stable conformation, while DIM2 and NOB1 are absent in this post-cleavage particle. Despite its overall impaired ATPase activity, the catalytic pocket of the RIO1-D324A mutant is occupied by ADP+pAsp341 (*Figure 3c*, *Figure 3—figure supplement 2*). This suggests that transition of pre-40S particles to state B involves ATP binding as well as the ATP hydrolysis activity of RIO-D324A, while trapping of ADP in the catalytic site prevents release from the matured particle. The structure of RIO1 was shown to switch to a so-called 'active' form when complexed either with ATP or with its hydrolysis products ADP/pAsp (*Ferreira-Cerca et al., 2014*). ATP fixation in the catalytic pocket of RIO1 might trigger its stabilization on the head of the pre-40S particle and bring RIO1 to contact DIM2. Furthermore, our data reveal a possible steric clash between the C-terminal domains of RIO1 and DIM2 (*Figure 6—figure supplement 1*). This suggests that RIO1 stabilization on pre-40S particles initiates DIM2 dislodging to uncover site 3 and allows rRNA cleavage by NOB1. Accordingly, our in vitro assays show that binding of ATP or AMP-PNP to RIO1 stimulates cleavage by NOB1 in human pre-40S particles (*Figure 6*), as concluded before in yeast (*Ferreira-Cerca et al., 2014*). Also supporting this scenario, recent in vitro studies in yeast have

shown that Rio1 forms a trimeric complex with Dim2 and Nob1 in the presence of AMP-PNP and not ADP (*Parker et al., 2019*). In addition, overexpression of Rio1 in yeast is sufficient to displace a catalytically inactive form of Nob1 from pre-40S particles (*Parker et al., 2019*). The recent study by *Ameismeier et al., 2020* reached similar conclusions regarding the role of RIO1. It also uncovered the role of EIF1AD as a new ribosome biogenesis factor. The structures published in this study suggest that both factors may functionally interact upstream of RPS26 incorporation.

Nevertheless, the action of RIO1 is not sufficient to efficiently trigger 18S-E pre-rRNA cleavage together with DIM2 and NOB1 dissociation from pre-40S particles, as shown by the blocking of RIO1-StHA pre-40S particles in the pre-cleavage state upon RPS26 knockdown. RPS26 is the only component of the mature small ribosomal subunit missing in state A. Its binding site overlaps with that of DIM2, and its final positioning on the 40S subunits is thus directly dependent on the release of DIM2. Importantly, addition of RPS26 was sufficient to stimulate in vitro cleavage of the 18S-E pre-rRNA in RPS26 depleted pre-40S particles, and it increased cleavage efficiency in the presence of ATP. While establishment of the full interaction of RIO1 with the head domain may provide a driving force to displace DIM2, RPS26 may potentiate its activity by competing with DIM2 binding. We propose a model in which displacement of DIM2 upon RIO1 binding in its final position (corresponding to state B) triggers the recruitment of RPS26 to the pre-40S particle, which in turn further displaces DIM2 from cleavage site 3. This cooperative action of RIO1 and RPS26 would then lead to a putative, probably short-lived cleavage-competent state (*Figure 7*), in which NOB1 can reach its substrate and process the 3′-end of the 18S rRNA. The status of ATP hydrolysis within RIO1 in this state is not known, and thus marked as 'ATP*' in *Figure 7*. The release of ADP and dephosphorylation of phospho-aspartate pAsp341 would then trigger a conformational change within RIO1 allowing its dissociation from pre-40S particles, as proposed for yeast Rio1 as well as other RIO kinases family members (*Ferreira-Cerca et al., 2014*; *Ferreira-Cerca et al., 2012*; *Knüppel et al., 2018*; *Turowski et al., 2014*). Along this view, RIO1 and RPS26 function as the two keys of a dual key lock that ensures a strict control of the ultimate steps of 40S subunit maturation before entry in translation.

## Human 40S subunit precursors do not form 80S-like particles

In yeast, expression of the Rio1-D244A mutant (equivalent to human RIO1-D324A) was shown to promote the accumulation of 80S-like particles, with which it co-purifies. In these 80S-like particles, pre-40S particles containing unprocessed 20S pre-rRNA, the last precursor to the 18S rRNA in yeast, are associated with large ribosomal subunits (*Ferreira-Cerca et al., 2014*; *Turowski et al., 2014*). Such particles were also observed with a number of other mutations targeting late-acting small ribosomal subunit biogenesis factors (*Lebaron et al., 2012*; *Parker et al., 2019*; *Scaiola et al., 2018*; *Strunk et al., 2012*) and were proposed to host the cleavage of the 20S pre-rRNA. Association of late pre-40S particles with large ribosomal subunits might serve as a checkpoint to verify that they are translation-competent after their final maturation (*Lebaron et al., 2012*; *Strunk et al., 2012*; *Turowski et al., 2014*). In addition, yeast 80S-like particles containing 20S pre-rRNA have been shown to be able to associate with polysomes and thus enter into the translating pool of ribosomes (*Belhabich-Baumas et al., 2017*; *Parker et al., 2019*; *Soudet et al., 2010*), suggesting some permissiveness between final pre-40S maturation and translation initiation events. In stark contrast, we found little evidence that human RIO1(kd)-StHA pre-ribosomal particles strongly associate with 60S subunits, neither by proteomics and northern blot, nor by cryo-EM single-particle analysis. Our results rather suggest that final rRNA maturation can occur within free pre-40S particles. Though ribosome biogenesis appears globally conserved within eukaryotic species, substantial differences in the pre-rRNA processing pathways, as well as in the dynamics of RBF association with pre-ribosomes in mammals and yeast, have been demonstrated (for review, *Cerezo et al., 2019*; *Henras et al., 2015*). The absence of 80S-like particles in late human pre-40S particles might constitute a significant difference in the maturation pathway of human pre-ribosomal particles compared to yeast.

## RIO1 probes tRNA translocation capacities of pre-40S particles

The function of RIO1 in the release of DIM2 and NOB1 echoes the roles that other NTPases play in pre-ribosomal particle remodeling along the maturation pathway. Their nucleotide-hydrolyzing activity was shown to power the release of other RBFs, as well as their own release, contributing to the

unidirectionality of the maturation (*Hedges et al., 2005*; *Kargas et al., 2019*; *Knüppel et al., 2018*; *Micic et al., 2020*; *Mitterer et al., 2019*; *Weis et al., 2015*; *Zemp et al., 2009*). NTPases might also have additional roles than conformational switches and probe the correct conformation of functional sites. For example, proper formation of the peptidyl-transferase center (PTC) in the large subunit may be checked by Nog2 and adjacent GTPases Nog1 and Nug1, which binds at or adjacent to the PTC (discussed by *Klinge and Woolford, 2019*). RIO1 binds to the same place as RIO2, which prevents premature engagement of pre-40S particles in translation by physically obstructing the mRNA groove and tRNA interaction sites (*Ameismeier et al., 2018*; *Heuer et al., 2017*; *Larburu et al., 2016*; *Scaiola et al., 2018*; *Strunk et al., 2012*). Furthermore, in the post-cleavage state B herein described, 18S rRNA G1639 appears to lock the RIO1(kd) catalytic pocket in its 'active' conformation (*Ferreira-Cerca et al., 2014*) through a pi-stacking interaction with phenylalanine F328 (*Figure 3c*). Such an interaction does not seem to be established with RIO2, whose catalytic pocket is more distant from this guanosine (*Ameismeier et al., 2018*). The N7-methylated G1639 is universally conserved and is involved in the correct positioning of tRNA in the P-site as well as its transfer to the E-site thanks to a switch mechanism (*Malygin et al., 2013*; *Selmer et al., 2006*). Through its cycle of association, ATPase hydrolysis, and subsequent dissociation from pre-40S particles, RIO1 might thus act as a conformational and functional probe that assesses the tRNA translocation capacities of the small subunit and couples triggering of the last maturation step to functional proofreading of the ribosome.

# Materials and methods

## Key resources table

| Reagent type (species) or resource | Designation | Source or reference | Identifiers | Additional information |
|---|---|---|---|---|
| Cell line (*Homo sapiens*) | HEK293 FlpIn T-REx HASt-DIM2 | (*Wyler et al., 2011*) DOI: 10.1261/rna.2325911 | | |
| Cell line (*Homo sapiens*) | HEK293 FlpIn T-REx HASt-LTV1 | (*Wyler et al., 2011*) DOI: 10.1261/rna.2325911 | | |
| Cell line (*Homo sapiens*) | HEK293 FlpIn T-REx RIOK1-StHA | (*Widmann et al., 2012*) DOI: 10.1091/mbc.E11-07-0639 | | |
| Cell line (*Homo sapiens*) | HEK293 FlpIn T-REx RIOK1 (D324A)-StHA | (*Widmann et al., 2012*) DOI: 10.1091/mbc. E11-07-0639 | | |
| Transfected construct (human) | si-RPS26 | Eurogentec | | 5'-GGACAAGGCCA UUAAGAAA dTdT-3' |
| Transfected construct (human) | si-EIF1AD | Eurogentec | | 5'-ACCGCAGAC AGUAUCAUGAGA-3' |
| Transfected construct (human) | si-control | Eurogentec | SR-CL000-005 | |
| Antibody | Anti-DIM2 (rabbit polyclonal) | (*Zemp et al., 2009*) DOI: 10.1083/jcb. 200904048 | | WB(1:2000) |
| Antibody | Anti-ENP1 (rabbit polyclonal) | (*Zemp et al., 2009*) DOI: 10.1083/jcb. 200904048 | | WB(1:5000) |
| Antibody | Anti-HA-HRP (rabbit polyclonal) | Sigma-Roche Applied Science | Cat# MMS-101P RRID:AB_2314672 | WB(1:1000) |

*Continued on next page*

*Continued*

| Reagent type (species) or resource | Designation | Source or reference | Identifiers | Additional information |
|---|---|---|---|---|
| Antibody | Anti-NOB1 (rabbit polyclonal) | (*Zemp et al., 2009*) DOI: 10.1083/jcb.200904048 | | WB(1:5000) |
| Antibody | Anti-RIOK2 (rabbit polyclonal) | (*Zemp et al., 2009*) DOI: 10.1083/jcb.200904048 | | WB(1:5000) |
| Antibody | Anti-RPS26 (rabbit polyclonal) | Genetex | GTX131193-S | WB(1:1000) |
| Antibody | Anti-RPS19 (rabbit polyclonal) | (This article) | | WB(1:5000) |
| Antibody | Anti-EIF1AD (rabbit polyclonal) | Proteintech | 20528–1-AP | WB(1:2000) |
| Sequence-based reagent | 3′18S probe | (*Larburu et al., 2016*) DOI: 10.1093/nar/gkw714 | | Northern blot probe, 5′-GATCCTTCCG CAGGTTCACCTACG-3′ |
| Sequence-based reagent | RNase H_3_Hyb1 (RNA/DNA/RNA) | Eurogentec | | 5′-UGUUACGAC UUUUACTTCCUCU AGAUAGUCAAGUUC-3′ |
| Sequence-based reagent | Lad_S oligo | Eurogentec | | 5′-TAATACGACTCA CTATAGGCGTAGG TGAACCTGCGGAA GGATCATTAACGG AGCCCGGAGGGCG AGGGATGAAGATG ATGAGCTCGGCAG GTCCTGAGGAGTGATGA-3′ |
| Sequence-based reagent | Lad_AS oligo | Eurogentec | | 5′- AAGGTGAATCA GCACTCAAGATCCT CATCACTCCTCAGGACC-3′ |
| Sequence-based reagent | Ladder sequence | This article | | 5′-UAAUACGACUCAC UAUAGGCGUAGG UGAACCUGCGGA AGGAUCAUUAACG GAGCCCGGAGGGC GAGGGAUGAAGAUG AUGAGCUCGGCAGG UCCUGAGGAGUGAU GAGGAUCUUGAGUG CUGAUUCACCUU-3′ |

## Cell culture and treatment with siRNA duplexes

HEK293 cells were cultured in Dulbecco's modified Eagle's medium supplemented with 10% fetal bovine serum and 1 mM sodium pyruvate. HEK293 FlpIn TRex cell lines (Invitrogen, RRID:CVCL_U427) and expressing HASt-DIM2, HASt-LTV1, RIO1(wt)-StHA, and RIO1(D324A)-StHA have been described previously (*Wyler et al., 2011*). Cell lines used in this study were not further authenticated after obtaining them from the indicated source. All cell lines were tested negative for mycoplasma using MycoAlert test (Lonza). None of the cell lines used in this study were included in the list of commonly misidentified cell lines maintained by International Cell Line Authentication Committee.

To knockdown expression of the corresponding human protein genes, siRNA duplex of RPS26 (5′-GGACAAGGCCAUUAAGAAAdTdT-3′) or EIF1AD (5′-ACCGCAGACAGUAUCAUGAGA-3′) (Eurogentec) was added at a final concentration of 500 nM to 100 µL of cell suspension (50 × 10^6 cells/mL diluted in Na phosphate buffer, pH 7.25, containing 250 mM sucrose and 1 mM MgCl_2). After electrotransformation at 240 V, cells were plated and collected 48 hr later. Control cells were electro-transformed with a scramble siRNA. Knockdown efficiency of siRNAs was assessed by quantitative PCR, using HPRT1 as an internal control.

## TAP purification of human pre-40S particles

To purify pre-ribosomal particles, the protocol described in *Wyler et al., 2011* was adapted as follows: expression of N-terminally HASt-tagged or C-terminally StHA-tagged bait proteins in stable HEK293 cells was induced with tetracycline (0.5 µg/mL) 24 hr prior to harvest. Cells were detached with PBS containing 0.5 mM EDTA and lysed in lysis buffer (10 mM Tris–HCl, pH 7.6, 100 mM KCl, 2 mM MgCl$_2$, 1 mM DTT, 0.5% NP-40, supplemented with protease and phosphatase inhibitors) using a dounce homogenizer. Lysed cells were centrifuged (4500 g, 12 min), and the lysate was incubated with EZview Red Anti-HA Affinity Gel (Sigma-Aldrich) for 2 hr in an overhead shaker. For electron microscopy studies, beads were washed six times with TAP buffer (20 mM Tris–HCl, pH 7.6, 100 mM KCl, 2 mM MgCl$_2$, 1 mM DTT) and eluted by incubation in TAP buffer supplemented with 0.2 mg/mL HA peptide (Sigma-Aldrich). Eluates were washed and concentrated by Vivacon 2, 100 kDa MWCO centrifugal devices (Sartorius). For subsequent analyses by silver staining, western blotting, and mass spectrometry, beads were washed three times with TAP buffer and bound material was eluted with SDS sample buffer.

## Protein digestion and nano-LC–MS/MS analysis

### Analysis of RIO1(kd)-StHA particles composition

Following TAP purification, 100 µL of concentrated RIO1 pre-40S particles, corresponding to 3.5 µg of protein, was reduced by incubation for 5 min at 95°C with 5 µL of Laemmli buffer containing 30 mM DTT and then alkylated with 90 mM iodoacetamide for 30 min at room temperature in the dark. The reduced/alkylated sample was then loaded onto SDS–PAGE gel (stacking 4% and separating 12% acrylamide). For one-shot analysis of the entire mixture, no fractionation was performed, and the electrophoretic migration was stopped as soon as the protein sample entered the separating gel as one single band. The proteins, revealed with Instant Blue (Expedeon) for 20 min, were found in one blue band of around 5 mm width. Single slice containing the whole sample was cut and washed before the in-gel digestion of the proteins overnight at 37°C with a solution of modified trypsin (sequence grade, Promega, Charbonnières, France). The resulting peptides were extracted from the gel by one round of incubation (15 min, 37°C) in 1% formic acid–acetonitrile (40%) and two rounds of incubation (15 min each, 37°C) in 1% formic acid–acetonitrile (1:1). The extracted fractions were air-dried. Tryptic peptides were resuspended in 14 µL of 2% acetonitrile and 0.05% trifluoroacetic acid (TFA). Nano-LC-MS/MS analysis was performed in duplicate injections using an Ultimate 3000 nanoRS system (Dionex, Amsterdam, The Netherlands) coupled with an LTQ-Orbitrap Velos mass spectrometer (Thermo Fisher Scientific, Bremen, Germany) operating in positive mode. Five microliters of each sample was loaded onto a C18-pre-column (300 µm inner diameter × 5 mm) at 20 µL/min in 2% ACN, 0.05% TFA. After 5 min of desalting, the pre-column was switched online with the analytical C18 nanocolumn (75 µm inner diameter × 15 cm, packed in-house) equilibrated in 95% solvent A (5% ACN, 0.2% FA) and 5% solvent B (80% ACN, 0.2% FA). Peptides were eluted by using a 5–50% gradient of solvent B for 105 min, at a flow rate of 300 nL/min. The LTQ-Orbitrap Velos was operated in data-dependent acquisition mode with the XCalibur software. Survey scans MS were acquired in the Orbitrap, on the 300–2000 m/z (mass to charge ratio) range, with the resolution set to a value of 60,000 at m/z 400. Up to 20 of the most intense multiply charged ions (>2+) per survey scan were selected for CID (collision-induced dissociation) fragmentation, and the resulting fragments were analyzed in the linear ion trap (LTQ). Dynamic exclusion was used within 60 s to prevent repetitive selection of the same peptide.

### Analysis of RIO1(kd)-StHA particles composition upon RPS26 depletion

Each protein sample corresponding to RIO1(kd)-StHA pre-40S particles purified from RPS26 depleted cells and control cells (64 and 94 µg of proteins in 300 µL of 40 mM Tris–HCl, 4% SDS, respectively) was reduced with 6 µL of 1 M DTT (20 mM final) during 10 min at 95°C under agitation, cooled to room temperature, and alkylated with 19 µL of 1 M iodoacetamide (60 mM final) during 30 min in the dark. Then, each sample was digested with trypsin using the S-Trap Micro spin column protocol (Protifi, Huntington, NY) according to the manufacturer's instructions (*HaileMariam et al., 2018*). The pooled digested peptide extracts were dried down and resuspended with 0.05% TFA in 2% ACN at a concentration of 3.8 µg/µL, vortexed, and sonicated for 10 min before injection. Peptides samples were analyzed in triplicate injections using an UltiMate 3000 RSLCnano LC system

(ThermoScientific, Dionex) coupled with an Orbitrap Fusion Tribrid mass spectrometer (Thermo Scientific, Bremen, Germany) operating in positive mode. Five microliters of each sample were loaded onto a 300 µm ID × 5 mm PepMap C18 pre-column (Thermo Scientific, Dionex) at 20 µL/min in 2% ACN, 0.05% TFA. After 5 min of desalting, peptides were on-line separated on a 75 µm ID × 50 cm C18 column (in-house packed with Reprosil C18-AQ Pur 3 µm resin, Dr. Maisch; Proxeon Biosystems, Odense, Denmark) equilibrated in 95% of buffer A (0.2% formic acid [FA]), with a gradient of 5% to 25% of buffer B (80% ACN, 0.2% FA) for 75 min and then 25–50% for 30 min at a flow rate of 300 nl/min. The instrument was operated in data-dependent acquisition mode using a top-speed approach (cycle time of 3 s). Survey scans MS were acquired in the Orbitrap over 400–2000 m/z with a resolution of 120,000 (at 200 m/z), an automatic gain control (AGC) target value of 4e5, and a maximum injection time of 50 ms. Most intense multiply charged ions (2 + to 7+) per survey scan were selected at 1.6 m/z with quadrupole and fragmented by higher energy collisional dissociation (HCD). The monoisotopic precursor selection was turned on, the intensity threshold for fragmentation was set to 50,000, and the normalized collision energy was set to 35%. The resulting fragments were analyzed in the Orbitrap with a resolution of 30,000 (at 200 m/z), an AGC target value of 5e4, and a maximum injection time of 60 ms. Dynamic exclusion was used within 60 s with a 10 ppm tolerance, to prevent repetitive selection of the same peptide. For internal calibration the 445.120025 ion was used as lock mass.

## Bioinformatic MS data analysis

Mascot (Mascot server v2.6.1; http://www.matrixscience.com) database search engine was used for peptide and protein identification using automatic decoy database search to calculate a false discovery rate (FDR). MS/MS spectra were compared to the SwissProt *H. sapiens* database (supplemented with the sequence of the D324A RIO1 mutant). MS mass tolerance was set at 5 ppm and MS/MS mass tolerance was set at 0.8 Da and 0.02 Da for data acquired on LTQ-Orbitrap Velos and Orbitrap Fusion Tribrid mass spectrometers, respectively. The enzyme selectivity was set to full trypsin with two missed cleavages allowed. Protein modifications were fixed carbamidomethylation of cysteines, variable phosphorylation of serine, threonine and tyrosine, variable oxidation of methionine, variable acetylation of protein N-terminus. Proline software (*Bouyssié et al., 2020*) was used for the validation and the label-free quantification of identified proteins in each sample. Mascot identification results were imported into Proline. Search results were validated with a peptide rank = 1 and at 1% FDR both at PSM level (on adjusted e-value criterion) and protein set level (on modified Mudpit score criterion). Label-free quantification was performed for all proteins identified: peptides were quantified by extraction of MS signals in the corresponding raw files, and post-processing steps were applied to filter, normalize, and compute protein intensities. For analysis of RIO1(kd)-StHA pre-40S particles composition, protein abundances were summarized in iBAQ values by dividing the protein intensities by the number of observable peptides (*Schwanhäusser et al., 2011*). Proteins with a ratio of observed peptide sequences over observable peptide sequences inferior to 30% were filtered out. Additionally, only the two most intense iBAQ logs (1E6 to 1E8) were represented in *Figure 1*, *Figure 1—figure supplement 2*. For analysis of RIO1(kd)-StHA pre-40S particle composition upon RPS26 depletion, after log2-transformation of the data, an unpaired two-tailed Student's t-test was performed and proteins were considered significantly enriched when their absolute log2-transformed fold change was higher than two and their *p*-value lower than 0.01. A volcano plot was drawn to visualize significant protein abundance variations between the two compared conditions and represents −log10 (*p*-value) according to the log2 ratio (*Figure 5—figure supplement 1*).

## Grid preparation and cryo-EM images acquisition

Cryo-EM grids were prepared and systematically checked at METI, Toulouse. Immediately after glow discharge, 3.5 µL of purified hRIO1(kd)-StHA particles (with RNA concentrations of ~60 ng/µL as estimated by Nanodrop measurement) was deposited onto QUANTIFOIL holey carbon grids (R2/1, 300 mesh with a 2 nm continuous layer of carbon on top). Grids were plunge-frozen using a Leica EM-GP automat; temperature and humidity level of the loading chamber were maintained at 20°C and 95%, respectively. Excess solution was blotted with a Whatman filter paper no. 1 for 1.7–1.9 s, and grids were immediately plunged into liquid ethane (−183°C).

Images were recorded on a Titan Krios electron microscope (FEI, ThermoFisher Scientific) located at EMBL, Heidelberg, Germany. The cryo-electron microscope was operating at 300 kV and was equipped with a Gatan K2 summit direct electron detector using counting mode. Automatic image acquisition was performed with SerialEM, at a magnification corresponding to a calibrated pixel size of 1.04 Å and a total electron dose of 29.88 e-/Å2 over 28 frames. Nominal defocus values ranged from −0.8 μm to −2.8 μm.

## Single-particle analysis

Nine thousand four hundred and ninety-four stacks of frames were collected at EMBL. Frame stacks were aligned to correct for beam-induced motion using MotionCor2 (*Zheng et al., 2017*). Contrast transfer function (CTF) and defocus estimation were performed on the realigned stacks using CTFFIND4 (*Rohou and Grigorieff, 2015*). After selection upon CTF estimation quality, maximum resolution on their power spectra, and visual checking, 'good' micrographs were retained for further analysis. The 2,126,610 particles were automatically picked and then extracted in boxes of 384 × 384 pixels, using the RELION 3.0 Autopick option. All subsequent image analysis was performed using RELION 3.0 (*Zivanov et al., 2018*; *Figure 2—figure supplement 1*). A first 2D classification was performed (on particles images binned by a factor of 8) to sort out ill-picked and methylosome particles. The 1,287,445 remaining particles were binned by a factor of 4 and subjected to a 3D classification in five classes, using the 40S subunit atomic model derived from the human ribosomal 3D structure (PDB-ID 6EK0) (*Natchiar et al., 2018*) low-pass filtered to 60 Å, as initial reference. One class harbored full 40S morphology and good level of details. The 484,429 particles from this class were re-extracted without imposing any binning factor, and a consensus 3D structure was obtained using RELION's 3D auto-refine option, with an overall resolution of 2.9 Å for FSC = 0.143 according to gold-standard FSC procedure. Particles were then submitted to focused 3D classifications with signal subtraction around the platform domain, to remove information coming from the body and head of the pre-40S particles, according to *Scheres, 2016*. Two of the five 3D classes yielded reconstructions with clear features. The first one, hereafter named state A, contained 104,844 particles and was further auto-refined to 3.1 Å resolution according to gold-standard FSC procedure; the second one (state B) comprised 276,012 particles and was auto-refined to 2.9 Å resolution. Subsequently, particles corresponding to the states A and B 'platform only' classes were retrieved from the dataset without signal subtraction and submitted to auto-refinement, yielding maps of 'full' states A and B particles solved to 3.22 and 2.96 Å, respectively (*Figure 2—figure supplement 1b, c*). Because significant motion of the head and platform regions compared to the body was observed, multi-body refinement (*Nakane et al., 2018*) was performed for both states by dividing pre-40S particles in three main domains: body, head, and platform. Multi-body refinement of state A gave rise to body, head, and platform domains solved to 3.14, 3.17, and 3.30 Å, respectively, after post-processing, while that of state B yielded resolutions of 2.98, 2.96, and 2.98 Å for the body, head, and platform regions, respectively (*Figure 2—figure supplement 1c*). Local resolution of all cryo-EM maps was estimated using the ResMap software (*Kucukelbir et al., 2014*; *Figure 2—figure supplement 2*).

## Interpretation of cryo-EM maps

Atomic models of pre-40S particle (PDB-ID 6G51) (*Ameismeier et al., 2018*) or the mature 40S subunit (PDB-ID 6EK0) (*Natchiar et al., 2018*) were first fitted in the cryo-EM maps of interest as rigid body using the 'fit' command in UCSF Chimera (*Pettersen et al., 2004*). Body, head, and platforms domains were modeled separately in the post-processed multi-body 'bodies' maps and then adapted to the composite full maps to generate whole atomic models using UCSF Chimera and Coot (*Emsley et al., 2010*).

Final atomic models of states A (pre-cleavage) and B (post-cleavage) pre-40S particles were refined using REFMAC5 (*Murshudov et al., 2011*) and Phenix_RealSpace_Refine (*Adams et al., 2010*), with secondary structure restraints for proteins and RNA generated by ProSMART (*Nicholls et al., 2014*) and LIBG (*Brown et al., 2015*). Final model evaluation was done with MolProbity (*Chen et al., 2010*). Overfitting statistics were calculated by a random displacement of atoms in the model, followed by a refinement against one of the half-maps in REFMAC5, and Fourier shell

correlation curves were calculated between the volume from the atomic model and each of the half-maps in REFMAC5 (*Supplementary file 1*).

Maps and models visualization was done with Coot and UCSF Chimera; figures were created using UCSF Chimera and ChimeraX (*Goddard et al., 2018*).

## RNase H digestion assay and RNA analysis

For RNase H digestion assays, 250 ng of pre-40S purified RNAs were denatured at 95°C for 6 min with a RNA/DNA/RNA reverse probe hybridizing in the 3′-end of 18S rRNA (probe RNaseH_3_Hyb1: 5′-UGUUACGACUUUU<u>ACTTCC</u>UCUAGAUAGUCAAGUUC-3′; 0.5 µL at 100 µM). After annealing by cooling down to room temperature for 20 min, the reaction mixture was diluted to 30 µL with a reaction mix containing 1X RNase H reaction buffer, 25 µM DTT, 0.5 U/l RNasin (Promega), and 50 U RNase H (New England Biolabs), and incubated at 37°C for 30 min. The reaction was then blocked by addition of 0.3 M sodium acetate, pH 5.2 and 0.2 mM EDTA, and the RNAs were recovered by ethanol precipitation after phenol–chloroform–isoamylalcohol (25:24:1) extraction.

RNAs were then separated on a 12% polyacrylamide gel (19:1) in 1X TBE buffer containing 7 M urea. RNAs were transferred to Hybond N + nylon membrane (GE-Healthcare, Orsay, France) and crosslinked under UV light. Membrane pre-hybridization was performed at 45°C in 6X SSC (saline-sodium citrate), 5X Denhardt's solution, 0.5% SDS, and 0.9 mg/mL tRNA. The 5′-radiolabeled oligonucleotide probe 3′18S (5′-GATCCTTCCGCAGGTTCACCTACG-3′) was added after 1 hr and incubated overnight at 45°C. Membranes were washed twice for 10 min in 2X SSC and 0.1% SDS and once in 1X SSC and 0.1% SDS, and then exposed. Signals were acquired with a Typhoon Trio PhosphorImager and quantified using the ImageLab software.

The RNA ladder corresponds to a 131 nucleotides sequence comprising the 3′18S probe and the T7 promoter sequence (*Supplementary file 1*). Using the oligonucleotides Lad_S and Lad_AS (*Supplementary file 1*), the DNA template for the RNA ladder was produced by PCR. Then, RNA was synthesized by in vitro transcription of the PCR products using the T7 RNA polymerase (Promega kit). Then, alkaline hydrolysis was performed on 500 ng of the synthesized RNA incubated in 50 mM sodium carbonate, pH 10.3, 1 mM EDTA for 5 min at 95°C.

## Overexpression and purification of hRPS26-His

hRPS26-HisTag construct was cloned into pET29b vector using NdeI/KpnI (GenScript). *Escherichia coli* BL21-CodonPlus (DE3) strain carrying pET29b-hRPS26-HisTag was induced by addition of 0.5 mM IPTG and incubated O.N at 20°C. Cells were pelleted and lysed by sonication in 20 mM HEPES, 50 mM NaCl, 2 mM EDTA, 10% glycerol, 20 mM Imidazole, and (EDTA)-free protease inhibitor (cOmpleteTM, Roche). The extract was treated with 750 U of Benzonase 30 min at room temperature under agitation and then clarified by centrifugation and filtration (0.22 µm). The recombinant protein was purified from the soluble fractions using FPLC (fast protein liquid chromatography, Äkta-Basic, GE-Healthcare), by a nickel-attached HiTrap chelating column (HisTrap 1 mL, GE-Healthcare) with a linear gradient of 20–500 mM imidazole. The eluted fractions containing His-tagged proteins were pooled, dialyzed, and concentrated on Amicon 10 kDa MWCO centrifugal devices (Sartorius) with dialysis buffer (50 mM Tris–HCl [pH 7.5], 0.1% Triton, 1 mM DTT, 10% glycerol, 5 mM MgCl$_2$). The purity of the protein was finally checked by SDS–PAGE 12% followed either by Coomassie blue staining or by western blot (*Figure 6—figure supplement 2*).

### In vitro pre-ribosome maturation assay

The method of pre-40S particle purification described above was followed, and instead of elution, beads were then washed three times with 1 mL of TAP buffer. Most of the supernatant was discarded, and 50 µL of buffer X (50 mM Tris–HCl pH 7.5, 150 mM NaCl, 5 mM MgCl2, 1 mM DTT, 1% Triton, and 10% glycerol) added to the beads (suspended in approx. 20 µL of remaining TAP buffer) to reach a final volume of 70 µL. Nucleotides at a final concentration of 1 mM and/or 2 µg of hRPS26 were added when required. Reactions were incubated at 37°C for 1 hr, and RNAs were then immediately extracted with Tri Reagent.

## Acknowledgements

This work was funded by the Agence Nationale de la Recherche (ANR 16-CE11-0029), the CNRS, the University of Toulouse-Paul Sabatier, the Région Occitanie, the European Union (Fonds Européens de Développement Régional, FEDER), and Toulouse Métropole. UK received funding from the Swiss National Science Foundation (grant 31003A_166565 and the NCCR 'RNA and disease'). Cryo-EM image acquisition was performed on the METI facility (CBI, Toulouse) and at the EMBL in Heidelberg with financial support from the iNext European initiative. Cryo-EM image analysis was performed using HPC resources from CALMIP (Grant 2018-[P1406]). We would like to thank the engineers and staff working on these facilities for their great help, as well as Marion Aguirrebengoa for her help on statistical analysis and Odile Burlet-Schiltz for her support.

## Additional information

### Funding

| Funder | Grant reference number | Author |
|---|---|---|
| Agence Nationale de la Recherche | 16-CE11-0029, ANR RIBOMAN | Laura Plassart<br>Ramtin Shayan<br>Natacha Larburu<br>Simon Lebaron<br>Marie-Françoise O'Donohue<br>Pierre-Emmanuel Gleizes<br>Celia Plisson-Chastang |
| Swiss National Science Foundation | 31003A_166565 | Christian Montellese<br>Ulrike Kutay |
| CALMIP | 2018-[P1406]) | Celia Plisson-Chastang |
| iNEXT | PID3684 - VID 5784 | Celia Plisson-Chastang |

The funders had no role in study design, data collection and interpretation, or the decision to submit the work for publication.

### Author contributions

Laura Plassart, Conceptualization, Data curation, Validation, Investigation, Visualization, Methodology, Writing - original draft, Writing - review and editing; Ramtin Shayan, Formal analysis, Investigation, Writing - original draft, Writing - review and editing; Christian Montellese, Investigation, Methodology, Writing - review and editing; Dana Rinaldi, Investigation, Methodology, Writing - original draft, Writing - review and editing; Natacha Larburu, Formal analysis, Investigation, Writing - review and editing; Carole Pichereaux, Formal analysis, Investigation; Carine Froment, Data curation, Formal analysis, Investigation, Methodology, Writing - review and editing; Simon Lebaron, Marie-Françoise O'Donohue, Conceptualization, Methodology, Writing - review and editing; Ulrike Kutay, Conceptualization, Resources, Funding acquisition, Methodology, Writing - review and editing; Julien Marcoux, Formal analysis, Supervision, Funding acquisition, Visualization, Methodology, Writing - original draft, Writing - review and editing; Pierre-Emmanuel Gleizes, Conceptualization, Formal analysis, Supervision, Funding acquisition, Methodology, Writing - original draft, Project administration, Writing - review and editing; Celia Plisson-Chastang, Conceptualization, Data curation, Formal analysis, Supervision, Funding acquisition, Investigation, Visualization, Methodology, Writing - original draft, Project administration, Writing - review and editing

### Author ORCIDs

Ulrike Kutay http://orcid.org/0000-0002-8257-7465
Julien Marcoux http://orcid.org/0000-0001-7321-7436
Pierre-Emmanuel Gleizes https://orcid.org/0000-0003-0830-7341
Celia Plisson-Chastang https://orcid.org/0000-0002-8439-8428

### Decision letter and Author response

Decision letter https://doi.org/10.7554/eLife.61254.sa1

Author response https://doi.org/10.7554/eLife.61254.sa2

## Additional files

### Supplementary files
• Supplementary file 1. Cryo-EM data collection, atomic models refinement, and validation statistics.

• Transparent reporting form

### Data availability

Mass spectrometry proteomics data have been deposited to the ProteomeXchange Consortium via the PRIDE partner repository with the dataset identifier PXD019270. Cryo-EM maps have been deposited in the Electron Microscopy Data Bank (EMDB), under the accession codes : EMD-11440 (State A multi-body composite map); EMD-11441 (State B multi-body composite map); EMD-11446 (State A, head); EMD-11445 (State A, body); EMD-11447 (State A, platform); EMD-11443 (State B, head); EMD-11442 (State B, body); EMD-11444 (State B, platform). Atomic coordinate models of State A and State B RIO1(kd)-StHA pre-40S particles have been deposited in the Protein Data Bank (PDB), with respective PDB accession codes 6ZUO and 6ZV6.

The following datasets were generated:

| Author(s) | Year | Dataset title | Dataset URL | Database and Identifier |
|---|---|---|---|---|
| Plassart L | 2021 | State A Multibody composite map | https://www.emdatare-source.org/EMD-11440 | EMDB, EMD-11440 |
| Plassart L | 2021 | State B Multibody composite map | https://www.emdatare-source.org/EMD-11441 | EMDB, EMD-11441 |
| Plassart L | 2021 | State A, head | https://www.emdatare-source.org/EMD-11446 | EMDB, EMD-11446 |
| Plassart L | 2021 | State A, body | https://www.emdatare-source.org/EMD-11445 | EMDB, EMD-11445 |
| Plassart L | 2021 | State A, platform | https://www.emdatare-source.org/EMD-11447 | EMDB, EMD-11447 |
| Plassart L | 2021 | State B, head | https://www.emdatare-source.org/EMD-11443 | EMDB, EMD-11443 |
| Plassart L | 2021 | State B, body | https://www.emdatare-source.org/EMD-11442 | EMDB, EMD-11442 |
| Plassart L | 2021 | State B, platform | https://www.emdatare-source.org/EMD-11444 | EMDB, EMD-11444 |
| Plassart L | 2021 | Human Pre-40S, State A | https://www.wwpdb.org/pdb?id=pdb_00006zuo | PDB, 6ZUO |
| Plassart L | 2021 | Human Pre-40S, State B | https://www.wwpdb.org/pdb?id=pdb_00006zv6 | PDB, 6ZV6 |
| Pichereaux C | 2021 | human late pre-40S particles (RIO1 (kd)-StHA | https://www.ebi.ac.uk/pride/archive/projects/PXD019270 | PRIDE, PXD019270 |

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
