## [Decision Letter]

**Acceptance summary:**

This study is significant in providing high-resolution cryo-EM images of the final two steps of 40S biogenesis in human cells, which provide structural insights into the roles of ribosome biogenesis factor Dim2 in blocking cleavage of the pre-rRNA by the endonuclease Nob1, and of the kinase Rio1 and 40S subunit protein Rps26 in displacing Dim2 to allow 18S rRNA cleavage by Nob1. Rps26 knockdown experiments add important biochemical evidence that Rps26 is critical for 18S rRNA cleavage and that it also enhances dissociation of Dim2 and Nob1; and in vitro cleavage experiments reconstitute the stimulatory function of Rp26 in 18S rRNA processing and confirm that ATP binding to Rio1 is sufficient to stimulate 18S rRNA cleavage by Nob1. The work also highlights a difference between human and yeast, as Nob1 is shown here to function on 40S subunits rather than on 80S ribosomes.

**Decision letter after peer review:**

Thank you for submitting your article "The final step of 40S ribosomal subunit maturation is controlled by a dual key lock" for consideration by *eLife*. Your article has been reviewed by 3 peer reviewers, one of whom is a member of our Board of Reviewing Editors, and the evaluation has been overseen by Cynthia Wolberger as the Senior Editor. The following individuals involved in review of your submission have agreed to reveal their identity: Arlen W Johnson (Reviewer #2); John Woolford (Reviewer #3).

The reviewers have discussed the reviews with one another and the Reviewing Editor has drafted this decision to help you prepare a revised submission.

Summary:

This paper presents cryo-EM structures of immature 40S subunits affinity-purified with a catalytically defective form of Rio1, in two major states. State A contains Nob1 and Dim2 but lacks Rps26 as well as density for the mutant Rio1-kd protein itself; and it has unprocessed 18S rRNA. Dim2 is seen to interact with the site of Nob1 cleavage in this state. State B lacks Nob1 and Dim2 but contains both Rps26 and Rio1-kd, and the 18S rRNA is processed. These findings suggest that these structures represent the final two steps in 18S rRNA processing, which appear to occur on 40S pre-ribosomes rather than in 80S complexes as seen in yeast. Comparing the two structures predicts a steric clash between Rio1 and Dim2, which might initiate Dim2 dissociation and attendant Rps26 binding to replace Dim2, as Rps26 occupies the same position on the 40S as Dim2. A role for Rps26 in stimulating 18S rRNA cleavage, presumably by preventing Dim2 association and steric hindrance of Nob1 access to the cleavage site, was provided by si-RNA knockdown of Rps26, which decreases the cleavage of 18S rRNA, while increasing occupancy of Nob1 and Dim2 in the pre-40S particles. In state B, Rio1-kd is bound to ADP and its Asp-D341 carries a phosphate group, suggesting that the kd mutation does not prevent ATP binding and hydrolysis, but rather release of the products of ATP hydrolysis. Supporting this inference, they show that adding ATP or non-hydolyzable AMP-PNP confers similar amounts of 18S rRNA cleavage in the pre-ribosomes affinity purified in association with Rio1-kd. Together, these results suggest the model that Rio1 and Rps26 function in a coordinated manner to displace Dim2 and enable access of Nob1 to the cleavage site, and that completion of ATP hydrolysis is necessary for release of Rio1 from the final, mature 40S subunit.

Essential revisions:

The reviewers had some concerns about the novelty of the main conclusions and the final model in Figure 7, which seemed to be fairly well supported by previous results already obtained in yeast; however, they agreed that it was important to address the mechanism in humans, and noted that the finding that cleavage apparently occurs on 40S rather than 80S pre-ribosomes represents an important difference with yeast. Obviously, providing data from high-resolution structures in support of the model is also quite valuable. At the same time, the reviewers felt that you should attempt to add more scientific value to the paper by performing a few additional experiments to support a direct, critical role for Rps26 in the proposed mechanism. This would entail (i) demonstrating that 40S particles affinity-purified with Rio1 from Rps26-repressed cells are defective for Nob1 cleavage compared to those purified from Rps26-replete cells, and (ii) attempt to restore Nob1 cleavage and evoke release of Dim2 and Nob1 to the Rps26-depleted particles by adding back purified Rps26.

Second, it is important to perform mass-spectrometry on the Rio1-associated 40S subunits following repression of Rps26 to insure that no other Rps proteins have been lost, in an effort to support the conclusion that Rps26 alone is required in conjunction with Rio1 for stimulating rRNA cleavage and dissociation of Dim2 and Nob1.

A third major point is to explain how you can differentiate between ADP/D341-P versus ATP in the EM density map.

Fourth, it is requested that you plot the cleavage data in a more straightforward and less misleading way, by providing the fraction cleaved (18S/(18S+18SE)).

Fifth, you should address whether the data in Figure 6A-B indicate significantly greater 18S rRNA cleavage in the 40S ribosomes containing WT Rio1 vs. the Rio1-kd mutant, which, if so, would seem to be at odds with your conclusion that ATP binding and hydrolysis, but not release of the hydrolysis products by Rio1 is sufficient for 18 rRNA cleavage.

Finally, you should attempt to revise the text to address all four major comments of Referee # 3; and also comment on whether your results can be reconciled with previous findings indicatingTsr2 loading of Rps26 in the nucleus-eg. might Rps26 associate with nuclear pre-40S particles and then undergo a rearrangement in the cytoplasm to help displace Dim2?

*Reviewer #1:*

Previous work has implicated NOB1 in the final processing of the 18S rRNA precursor in cytoplasmic 40S pre-ribosomes, and showed that DIM2 restricts this endonucleolytic cleavage activity. It was also shown previously that Rio1 and Rps26 are necessary for cleavage by Nob1, and for release of Dim2 and Nob1 from the mature 40S particle. This paper presents cryo-EM structures of 40S subunits affinity-purified with a catalytically defective form of Rio1, in two major states. State A contains Nob1 and Dim2 but lacks Rps26 as well as density for the mutant Rio1-kd protein itself; and it has unprocessed 18S rRNA. Dim2 is seen to interact with the site of Nob1 cleavage in this state. State B lacks Nob1 and Dim2 but contains both Rps26 and Rio1-kd, and the 18S rRNA is processed. These findings suggest that these structures represent the final two steps in 18S rRNA processing, which appear to occur on 40S pre-ribosomes rather than in 80S complexes as seen in yeast. Comparing the two structures predicts a steric clash between Rio1 and Dim2, which might initiate Dim2 dissociation and attendant Rps26 binding to replace Dim2, as Rps26 occupies the same position on the 40S as Dim2. A role for Rps26 in stimulating 18S rRNA cleavage, presumably by preventing Dim2 association and steric hindrance of Nob1 access to the cleavage site, was provided by si-RNA knockdown of Rps26, which decreases the cleavage of 18S rRNA, while increasing occupancy of Nob1 and Dim2 in the pre-40S particles. In state B, Rio1-kd is bound to ADP and its Asp-D341 carries a phosphate group, suggesting that the kd mutation does not prevent ATP binding and hydrolysis, but rather release of the products of ATP hydrolysis. Supporting this inference, they show that adding ATP or non-hydolyzable AMP-PNP confers similar amounts of 18S rRNA cleavage in the pre-ribosomes affinity purified in association with Rio1-kd. Together, these results suggest the model that Rio1 and Rps26 function in a coordinated manner to displace Dim2 and enable access of Nob1 to the cleavage site, and that completion of ATP hydrolysis is necessary for release of Rio1 from the final, mature 40S subunit.

The results appear to be significant in providing high resolution cryo-EM images of the likely final two steps of 40S biogenesis, and provide structural insights into the roles of Dim2 in blocking cleavage of the pre-rRNA by Nob1, and of Rio1 and Rps26 in displacing Dim2 to allow 18S rRNA cleavage by Nob1. The Rps26 knockdown experiments add further evidence supporting previous findings that Rps26 is important for 18S rRNA cleavage and provide evidence that it also enhances dissociation of Dim2 and Nob1; and the in vitro cleavage experiments support previous findings from yeast that ATP binding to Rio1 is sufficient for 18S rRNA cleavage by Nob1.

1. While the cryo-EM structures are certainly valuable, there is a concern that the major conclusions of the paper, embodied in the model in Figure 7, were already very likely to be true based on previous findings from yeast, cited in the Introduction, that Dim2 restricts endonucleolytic cleavage by Nob1 by masking the cleavage site, that Nob1 and Dim2 dissociate only after 18S rRNA cleavage, that Rio1 and Rps26 are both necessary for cleavage by Nob1, and that Rio1 is also necessary for the release of Dim2 and Nob1 from the mature 40S subunit.

2. It might be necessary to perform MS analysis on the Rio1-associated 40S subunits following knock-down of Rps26 to insure that no other Rps proteins are lost in order to bolster the conclusion that Rps26 alone is required in conjunction with Rio1 for stimulating rRNA cleavage and dissociation of Dim2 and Nob1.

3. Results in Figure 6A-B seem to indicate significantly greater 18S rRNA cleavage in the 40S ribosomes containing WT Rio1 vs. the Rio1-kd mutant, which seems at odds with the conclusion that ATP binding and hydrolysis, but not release of the hydrolysis products by Rio1 is sufficient for 18 rRNA cleavage. This needs to be addressed.

4. It may be important to show directly that Rps26 is required for 18S rRNA cleavage by carrying out in vitro cleavage assays like those shown in Figure 6 following knock-down of Rps26.

*Reviewer #2:*

Ribosome assembly is a complex pathway that is thought to involve quality control checks to ensure the correct assembly of ribosomes. In yeast, a major event in assessing the assembly of small subunits is the formation of an 80S-like complex to promote final cleavage of 18S rRNA in the cytoplasm. However, less is known about the pathway of ribosome assembly in humans. In this manuscript, the authors use cryo-EM to analyze particles affinity purified from human cells with a catalytic mutant of RIO1 kinase. The comparable mutant in yeast associates with 80S-like particles. The authors resolve two apparently sequential states of 40S maturation that represent intermediates before and after cleavage at site 3 to generate the mature 3'-end of 18S. Surprisingly, the authors find no evidence for an 80S-like particle. Consistent with earlier work from yeast, the authors show that cleavage at site 3 depends on RIO1 and is stimulated by ATP. Their data also strongly implicate the loading of RPS26 in displacement of DIM2 to allow cleavage by NOB1. A role for Rps26 in cleavage of pre-18S rRNA has also previously been reported in yeast. In general the data are clearly presented and the manuscript is well-written.

1. The authors present the model that RIO1, together with RPS26, displace DIM1 to promote cleavage at site 3 by NOB1. The authors have nearly all the tools to test this in vitro. Does the addition of RPS26 to particles affinity purified with RIO1 from RPS26-repressed cells promote NOB1 cleavage and release of DIM2?

2. In Figure 3C the authors suggest that RIO1 contains ADP and phospho D341. It is not clear from the figure that this is the case and not ATP. A supplemental figure showing the fit of ADP and P-D341 vs ATP in the density map would help make this point.

3. In Figures4C, 5B and 6B the data are not displayed appropriately. The authors plot the ratio of 18S/18SE. However, this misleadingly exaggerates the results: as the efficiency of cleavage approaches 100%, this ratio goes to infinity. The authors should plot this as fraction cleaved (18S/(18S+18SE)).

*Reviewer #3:*

During late stages of eukaryotic ribosome assembly the final step of pre-rRNA processing occurs and the last four ribosome biogenesis factors are released. Work described here provides a more detailed understanding of this process for human pre-40S particles: assembly factor DIM2 prevents access of the NOB1 endonuclease to its cleavage site. DIM2 is displaced by entry of RIO1 and RPS26, enabling cleavage of the 20S pre-rRNA by NOB1. A conformational change in RIO1 triggers its dissociation. Experiments in this well written manuscript provide strong support for this model, which extends our understanding of these steps from what had previously been found in yeast.

The manuscript could be improved by addressing the following:

1. In the Introduction, or at the beginning of Results, the authors might provide a little more information about RIO1 autocatalytic activity and its putative role, before describing its use as an affinity purification bait. Can they estimate how much activity remains in the D324A mutant that they use?

2. That the bait protein Rio1 cannot be visualized in State A particles might be confusing to some readers. The authors suggest that it is loosely bound and falls off during purification. They should provide a slightly more detailed explanation of when during purification this might occur…certainly after attachment to the affinity gel. It also seems reasonable that RIO1 is present but particularly flexible in state A particles.

3. Page 11: Is there any evidence for how RPS26 initially contacts binds to preribosomes? Might it bind near DIM2, then undergo a series of conformational shifts to help displace DIM2?

4. In my opinion, an interesting question in ribosome assembly is how is each assembly factors is released from preribosomes. There are more than 200 assembly factors , but many many fewer NTPases. Does each NTPase help many AFS depart or are there additional release mechanisms, or both? The authors might highlight this issue a little more, e.g., by comparing what they found with a few other examples from the literature, especially if there are any cases where NTPases are not yet known to be involved in an AF release. How does binding of one protein displace another?

---

## [Author Response]

Essential revisions:The reviewers had some concerns about the novelty of the main conclusions and the final model in Figure 7, which seemed to be fairly well supported by previous results already obtained in yeast; however, they agreed that it was important to address the mechanism in humans, and noted that the finding that cleavage apparently occurs on 40S rather than 80S pre-ribosomes represents an important difference with yeast. Obviously, providing data from high-resolution structures in support of the model is also quite valuable. At the same time, the reviewers felt that you should attempt to add more scientific value to the paper by performing a few additional experiments to support a direct, critical role for Rps26 in the proposed mechanism. This would entail:(i) Demonstrating that 40S particles affinity-purified with Rio1 from Rps26-repressed cells are defective for Nob1 cleavage compared to those purified from Rps26-replete cells.

As shown in Figure 5c, particles purified with tagged wild-type RIO1 are devoid of RPS26 even in cells expressing RPS26, and contain DIM2 and NOB1 like the particles purified from RPS26-depleted cells. The new mass spectrometry data provided here further show that the composition of pre-40S particles isolated from RPS26-depleted cells is not different from that purified in control cells, beyond the absence of RPS26. RPS26 appears to be the trigger of cleavage, and we could not isolate uncleaved particles containing RPS26. Similarly, Ameismeier et al. (2020) did not identify particles containing DIM2 and NOB1 together with RPS26. Therefore, we felt that the only way to directly show the function of RPS26 was to use the in vitro approach.

(ii) Attempt to restore Nob1 cleavage and evoke release of Dim2 and Nob1 to the Rps26-depleted particles by adding back purified Rps26.

As requested, we have performed in vitro cleavage experiments which show that adding back purified S26 to RIO1-containing/S26-depleted particles stimulates 18S-E cleavage. This effect is potentialized by the combined addition of ATP. We have modified the manuscript text accordingly: in the introduction section (last two sentences):

*“*in vitro, cleavage of the 18S-E pre-rRNA was partially stimulated by ATP binding to RIO1 in RPS26-depleted pre-40S particles, and was enhanced by adding back purified RPS26. These data suggest a model, in which ATP-bound RIO1 and RPS26 cooperatively displace DIM2 to activate the final cleavage of the 18S rRNA 3’ end by NOB1.”

In the Results section, p. 10, the Results section title was modified as follows:

“ATP binding by RIO1 and addition of RPS26 stimulate in vitro rRNA cleavage by NOB1 in RPS26-depleted pre-40S particles”, and within the section (p.10):

“We next sought to assess the role of RPS26 and produced recombinant human RPS26 (Figure 6—figure supplement 2).[…] These data show that in vitro cleavage by NOB1 in pre-40S-particle is stimulated both by RIO1 and RPS26 and support the hypothesis of a cooperative action of RPS26 and RIO1 in triggering the final step of 18S rRNA maturation..”

Second, it is important to perform mass-spectrometry on the Rio1-associated 40S subunits following repression of Rps26 to insure that no other Rps proteins have been lost, in an effort to support the conclusion that Rps26 alone is required in conjunction with Rio1 for stimulating rRNA cleavage and dissociation of Dim2 and Nob1.

We have performed the required proteomics experiments, and compared the proteic composition of RIO1(kd)-StHA purified particles in presence or in absence of RPS26. The corresponding figure is presented as Figure 5—figure supplement 1. Overall, this confirmed that RPS26 depletion does not affect the abundance levels of other RPS within RIO1(kd)-associated particles. However, we noticed that together with RPS26, EIF1AD was also significantly less purified upon RPS26 depletion, suggesting a putative regulation link between these two proteins. Since the purification bait, RIO1, is known to be associated with both methylosome and pre-40S particles, we have also monitored the level of EIF1AD within pre-40S particles by measuring the EIF1AD/RPS19 ratio on western blot experiments. This shows only a mild reduction (~25%) of EIF1AD within pre-40S particles upon RPS26 depletion.

To summarize these observations, we have added the following sentence to the Results section, p.9 : “However, upon RPS26 depletion, RIO1(wt)-StHA co-purified with late cytoplasmic pre-40S particles, as attested by the presence of NOB1 and DIM2 (Figure 5c), as well as that of 18S-E pre-rRNA (Figure 5a). These data indicate that RIO1 association to late cytoplasmic pre-40S particles is stabilized in the absence of RPS26, while release of NOB1 and DIM2 is inhibited.

“Comparative proteomics analyses of RIO1(kd)-StHA associated pre-40S particles in the absence or presence of RPS26 confirmed that no other RPS was lost upon RPS26 depletion (Figure 5—figure supplement 1). […] Altogether, these data lead us to conclude that RPS26 intervenes directly in the mechanism triggering rRNA cleavage by NOB1 at site 3 and dissociation of NOB1, DIM2 and RIO1 from pre-40S particles.”

A third major point is to explain how you can differentiate between ADP/D341-P versus ATP in the EM density map.

We have added a supplementary figure (Figure 3—figure supplement 2) showing the cryoEM map fitted with either ADP and P-D341 or ATP, which better illustrates this point.

Fourth, it is requested that you plot the cleavage data in a more straightforward and less misleading way, by providing the fraction cleaved (18S/(18S+18SE)).

All figures showing cleavage analysis have been modified according to this request.

Fifth, you should address whether the data in Figure 6A-B indicate significantly greater 18S rRNA cleavage in the 40S ribosomes containing WT Rio1 vs. the Rio1-kd mutant, which, if so, would seem to be at odds with your conclusion that ATP binding and hydrolysis, but not release of the hydrolysis products by Rio1 is sufficient for 18 rRNA cleavage.

We added a new series of cleavage experiments with more points to gain statistical power (figure 6c), and modified figure 6 to represent the fraction of cleaved rRNA as suggested above (18S/(18S+18SE)). Furthermore, we improved cleavage efficiency by using Mg^2+^ instead of Mn^2+^ as a divalent cation. The new plots do not indicate greater cleavage in pre-40S containing RIO1(wt) vs. the RIO1(kd) mutant in the presence of ATP (figure 6, b and c). We do observe however that cleavage is somewhat more efficient with ATP than with AMP-PNP in pre-40S particles containing wild-type RIO1 (Figure 6b). Such a difference is also visible for RIO1(kd) containing pre-40S particles, even though it fails to be statistically significant in these experiments (p=0.063 with a stringent paired non-parametric test). This result fits with our cryo-EM observation that the catalytic site of RIO1 contains ADP+Pi rather than ATP in the post-cleavage state, which strongly suggests that cleavage is favored by ATP hydrolysis. Overall, the results of these in vitro experiments do support our conclusion that ATP binding and hydrolysis is sufficient for 18S rRNA cleavage, while release of hydrolysis products appears to be dispensable (or at least not strictly required), there again consistent with the presence of ADP-Pi in RIO1 catalytic site in post-cleavage state B.

Finally, you should attempt to revise the text to address all four major comments of Referee # 3; and also comment on whether your results can be reconciled with previous findings indicatingTsr2 loading of Rps26 in the nucleus-eg. might Rps26 associate with nuclear pre-40S particles and then undergo a rearrangement in the cytoplasm to help displace Dim2?

In our opinion, the timing of association of RPS26 to pre-ribosomal particles is still a matter of debate in yeast, and to our knowledge, has never been studied in human cells. This chronology of association might differ between human and yeast. However we acknowledge that the work of Vikram Panse’s lab on the interplay between Rps26 and Tsr2 should clearly be discussed in this manuscript. Thus, in the introduction section (p.4) , we have added the following paragraph:

“Like RIO2, another ATPase of the same family that intervenes earlier in pre-40S particle maturation, RIO1 adopts different conformations depending on its nucleotide binding state (Ferreira-Cerca et al., 2014). […] One cannot exclude that association of RPS26 to nuclear precursors might be highly labile and becomes more stable only during later maturation steps, but evidence is still lacking to date for the presence of human RPS26 in early precursors.”

Reviewer #1:Previous work has implicated NOB1 in the final processing of the 18S rRNA precursor in cytoplasmic 40S pre-ribosomes, and showed that DIM2 restricts this endonucleolytic cleavage activity. It was also shown previously that Rio1 and Rps26 are necessary for cleavage by Nob1, and for release of Dim2 and Nob1 from the mature 40S particle. This paper presents cryo-EM structures of 40S subunits affinity-purified with a catalytically defective form of Rio1, in two major states. State A contains Nob1 and Dim2 but lacks Rps26 as well as density for the mutant Rio1-kd protein itself; and it has unprocessed 18S rRNA. Dim2 is seen to interact with the site of Nob1 cleavage in this state. State B lacks Nob1 and Dim2 but contains both Rps26 and Rio1-kd, and the 18S rRNA is processed. These findings suggest that these structures represent the final two steps in 18S rRNA processing, which appear to occur on 40S pre-ribosomes rather than in 80S complexes as seen in yeast. Comparing the two structures predicts a steric clash between Rio1 and Dim2, which might initiate Dim2 dissociation and attendant Rps26 binding to replace Dim2, as Rps26 occupies the same position on the 40S as Dim2. A role for Rps26 in stimulating 18S rRNA cleavage, presumably by preventing Dim2 association and steric hindrance of Nob1 access to the cleavage site, was provided by si-RNA knockdown of Rps26, which decreases the cleavage of 18S rRNA, while increasing occupancy of Nob1 and Dim2 in the pre-40S particles. In state B, Rio1-kd is bound to ADP and its Asp-D341 carries a phosphate group, suggesting that the kd mutation does not prevent ATP binding and hydrolysis, but rather release of the products of ATP hydrolysis. Supporting this inference, they show that adding ATP or non-hydolyzable AMP-PNP confers similar amounts of 18S rRNA cleavage in the pre-ribosomes affinity purified in association with Rio1-kd. Together, these results suggest the model that Rio1 and Rps26 function in a coordinated manner to displace Dim2 and enable access of Nob1 to the cleavage site, and that completion of ATP hydrolysis is necessary for release of Rio1 from the final, mature 40S subunit.The results appear to be significant in providing high resolution cryo-EM images of the likely final two steps of 40S biogenesis, and provide structural insights into the roles of Dim2 in blocking cleavage of the pre-rRNA by Nob1, and of Rio1 and Rps26 in displacing Dim2 to allow 18S rRNA cleavage by Nob1. The Rps26 knockdown experiments add further evidence supporting previous findings that Rps26 is important for 18S rRNA cleavage and provide evidence that it also enhances dissociation of Dim2 and Nob1; and the in vitro cleavage experiments support previous findings from yeast that ATP binding to Rio1 is sufficient for 18S rRNA cleavage by Nob1.1. While the cryo-EM structures are certainly valuable, there is a concern that the major conclusions of the paper, embodied in the model in Figure 7, were already very likely to be true based on previous findings from yeast, cited in the Introduction, that Dim2 restricts endonucleolytic cleavage by Nob1 by masking the cleavage site, that Nob1 and Dim2 dissociate only after 18S rRNA cleavage, that Rio1 and Rps26 are both necessary for cleavage by Nob1, and that Rio1 is also necessary for the release of Dim2 and Nob1 from the mature 40S subunit.

We and others have already shown significant differences between ribosome maturation processes in yeast and humans. The data reported here underline yet another divergence: as stated in the discussion, unlike in yeast we did not find any 80S-like particles despite purifying particles using the same bait. This suggests that the last maturation steps of the small ribosomal subunit differ between these two species, which calls for a specific characterization in human cells. Although RIO1 and RPS26 were both shown to be required in the processing of 18S-E pre-rRNA, a precise description of the underlying mechanism is still missing. To our knowledge this study is the first to show a direct and combined action of these two proteins in the cleavage of the 18S-E pre-rRNA by NOB1.

2. It might be necessary to perform MS analysis on the Rio1-associated 40S subunits following knock-down of Rps26 to insure that no other Rps proteins are lost in order to bolster the conclusion that Rps26 alone is required in conjunction with Rio1 for stimulating rRNA cleavage and dissociation of Dim2 and Nob1.

This analysis has been performed; please see above.

3. Results in Figure 6A-B seem to indicate significantly greater 18S rRNA cleavage in the 40S ribosomes containing WT Rio1 vs. the Rio1-kd mutant, which seems at odds with the conclusion that ATP binding and hydrolysis, but not release of the hydrolysis products by Rio1 is sufficient for 18 rRNA cleavage. This needs to be addressed.

We agree that our previous representation of in vitro cleavage analysis might have been misleading; we have now represented the fraction of cleaved rRNA as requested (18S/(18S+18SE)), and believe this new plotting lifts the ambiguity (see above).

4. It may be important to show directly that Rps26 is required for 18S rRNA cleavage by carrying out in vitro cleavage assays like those shown in Figure 6 following knock-down of Rps26.

We also performed the suggested experiments; see above.

Reviewer #2:Ribosome assembly is a complex pathway that is thought to involve quality control checks to ensure the correct assembly of ribosomes. In yeast, a major event in assessing the assembly of small subunits is the formation of an 80S-like complex to promote final cleavage of 18S rRNA in the cytoplasm. However, less is known about the pathway of ribosome assembly in humans. In this manuscript, the authors use cryo-EM to analyze particles affinity purified from human cells with a catalytic mutant of RIO1 kinase. The comparable mutant in yeast associates with 80S-like particles. The authors resolve two apparently sequential states of 40S maturation that represent intermediates before and after cleavage at site 3 to generate the mature 3'-end of 18S. Surprisingly, the authors find no evidence for an 80S-like particle. Consistent with earlier work from yeast, the authors show that cleavage at site 3 depends on RIO1 and is stimulated by ATP. Their data also strongly implicate the loading of RPS26 in displacement of DIM2 to allow cleavage by NOB1. A role for Rps26 in cleavage of pre-18S rRNA has also previously been reported in yeast. In general the data are clearly presented and the manuscript is well-written.1. The authors present the model that RIO1, together with RPS26, displace DIM1 to promote cleavage at site 3 by NOB1. The authors have nearly all the tools to test this in vitro. Does the addition of RPS26 to particles affinity purified with RIO1 from RPS26-repressed cells promote NOB1 cleavage and release of DIM2?

We have added purified S26 to late pre-40S particles and showed that indeed this potentialized rRNA cleavage, please see above.

2. In Figure 3C the authors suggest that RIO1 contains ADP and phospho D341. It is not clear from the figure that this is the case and not ATP. A supplemental figure showing the fit of ADP and P-D341 vs ATP in the density map would help make this point.

We have added a supplementary figure (Figure 3—figure supplement 2) showing the cryoEM map fitted with either ADP and P-D341 or ATP to prove this point.

3. In Figures 4C, 5B and 6B the data are not displayed appropriately. The authors plot the ratio of 18S/18SE. However, this misleadingly exaggerates the results: as the efficiency of cleavage approaches 100%, this ratio goes to infinity. The authors should plot this as fraction cleaved (18S/(18S+18SE)).

We have addressed this issue, please see above.

Reviewer #3:During late stages of eukaryotic ribosome assembly the final step of pre-rRNA processing occurs and the last four ribosome biogenesis factors are released. Work described here provides a more detailed understanding of this process for human pre-40S particles: assembly factor DIM2 prevents access of the NOB1 endonuclease to its cleavage site. DIM2 is displaced by entry of RIO1 and RPS26, enabling cleavage of the 20S pre-rRNA by NOB1. A conformational change in RIO1 triggers its dissociation. Experiments in this well written manuscript provide strong support for this model, which extends our understanding of these steps from what had previously been found in yeast.The manuscript could be improved by addressing the following:1. In the Introduction, or at the beginning of Results, the authors might provide a little more information about RIO1 autocatalytic activity and its putative role, before describing its use as an affinity purification bait. Can they estimate how much activity remains in the D324A mutant that they use?

As suggested, we have added several sentences to try to emphasize on the ATPase role of RIO1 in small ribosomal subunits maturation. In the introduction section, we have modified the following sentence (p.4) as follows:

“Absence of RIO1 or suppression of its catalytic activity impairs both rRNA cleavage and release of NOB1 and DIM2 in yeast and human cells (Ferreira-Cerca et al., 2014; Turowski 2014, Widmann et al., 2012), suggesting that RIO1 association to pre-40S particles as well as its ATPase activity are both required to yield fully functional small ribosomal subunits.”

Because we could not retrieve the original data, we could not precisely quantify RIO1 D324A autophosphorylation activity from our previous results (Widmann et al., 2012), so we based our estimation on *Chaetomium thermophilum* ATPase activity results presented in Ferreira-Cerca et al. (2014). Thus, in the first section of the results part (p.5), we have modified the first paragraph as follows:

“Autophosphorylation of this mutant in the presence of ATP was previously shown to be strongly impaired (Widmann et al., 2012), and free π release activity reduced by at least 50% in its *Chaetomium thermophilum* D281A ortholog (Ferreira-Cerca et al., 2014)”.

We hope these few sentences will be useful to the reader.

2. That the bait protein Rio1 cannot be visualized in State A particles might be confusing to some readers. The authors suggest that it is loosely bound and falls off during purification. They should provide a slightly more detailed explanation of when during purification this might occur…certainly after attachment to the affinity gel. It also seems reasonable that RIO1 is present but particularly flexible in state A particles.

We believe that the latter hypothesis is true; thus, in the Results sections describing the state A cryo-EM map (p. 6), we have added the following sentences:

“Furthermore, NOB1 and DIM2 are the only two RBFs that can clearly be distinguished, while RIO1(kd)-StHA, the protein used as purification bait, cannot be clearly positioned on this cryo-EM map. […] Indeed, while this manuscript was in revision, a cryo-EM study focusing on similarly late human pre-40S particles revealed significant changes in RIO1 position on maturing pre-40S particles (Ameismeier et al., 2020).”

3. Page 11: Is there any evidence for how RPS26 initially contacts binds to preribosomes? Might it bind near DIM2, then undergo a series of conformational shifts to help displace DIM2?

Beyond V. Panse’s lab work on early association of RPS26 to nucle(ol)ar yeast pre-40S particles, which has now been briefly described in the introduction section of our manuscript, there is to our knowledge no evidence regarding how and where RPS26 initially binds to pre-ribosomes. In their recent cryo-EM study, Ameismeier et al. only detected RPS26 in pre-40S particles in its final position, despite the depth of their analysis. This does not necessarily plead for a stepwise assembly; what reviewer 3 proposes might be true, but we have no structural or functional data to sustain this hypothesis in human cells.

4. In my opinion, an interesting question in ribosome assembly is how is each assembly factors is released from preribosomes. There are more than 200 assembly factors , but many many fewer NTPases. Does each NTPase help many AFS depart or are there additional release mechanisms, or both? The authors might highlight this issue a little more, e.g., by comparing what they found with a few other examples from the literature, especially if there are any cases where NTPases are not yet known to be involved in an AF release. How does binding of one protein displace another?

Thank you for this very interesting observation, which could be the object of a review article. Indeed, many NTPases are currently seen as “power switches'', that help release other assembly factors from maturing pre-ribosomes, and allow progress to later assembly steps of these ribosomal particles. We believe this is indeed the case, but also that NTPases might have other functions. For instance, our data show a very close interaction between RIO1 and a functional site of the small ribosomal subunit (the N7-methylated Guanosine 1639), which suggests that RIO1 might also directly probe the functionality of this nucleotide). In order to support this idea, we have added the following sentences to the final part of the discussion:

“**RIO1 probes tRNA translocation capacities of pre-40S particles''** (p.13) as follows:

**“**The function of RIO1 in the release of DIM2 and NOB1 echoes the roles that other NTPases play in pre-ribosomal particle remodeling along the maturation pathway. […] Furthermore, in the post-cleavage state B herein described […]…”